# Air-conducted ultrasound below the hearing threshold elicits functional changes in the cognitive control network

Markus Weichenberger[1]*, Marion U. Bug[2], Rüdiger Brühl[2], Bernd Ittermann[2], Christian Koch[2], Simone Kühn[1,3]

1 Max Planck Institute for Human Development, Lise Meitner Group for Environmental Neuroscience, Berlin, Germany, 2 Physikalisch-Technische Bundesanstalt (PTB), Berlin, Germany, 3 University Clinic Hamburg-Eppendorf, Clinic and Policlinic for Psychiatry and Psychotherapy, Hamburg, Germany

* markus.weichenberger@triaplus.ch

**Data Availability Statement:** Data are available on the Max Planck Institute for Human Development data server for researchers who meet the criteria for access to confidential data. These restrictions

## Abstract

Air-conducted ultrasound (> 17.8 kHz; US) is produced by an increasing number of technical devices in our daily environment. While several studies indicate that exposure to US in public spaces can lead to subjective symptoms such as 'annoyance' or 'difficulties in concentration', the effects of US on brain activity are poorly understood. In the present study, individual hearing thresholds (HT) for sounds in the US frequency spectrum were assessed in 21 normal-hearing participants. The effects of US were then investigated by means of functional magnetic resonance imaging (fMRI). 15 of these participants underwent three resting-state acquisitions, two with a 21.5 kHz tone presented monaurally at 5 dB above (ATC) and 10 dB below (BTC) the HT and one without auditory stimulation (NTC), as well as three runs of an n-back working memory task involving similar stimulus conditions (n-ATC, n-BTC, n-NTC). Comparing data gathered during n-NTC vs. fixation, we found that task performance was associated with the recruitment of regions within the cognitive control network, including prefrontal and parietal areas as well as the cerebellum. Direct contrasts of the two stimulus conditions (n-ATC & n-BTC) vs. n-NTC showed no significant differences in brain activity, irrespective of whether a whole-brain or a region of interest approach with primary auditory cortex as the seed was used. Likewise, no differences were found when the resting-state runs were compared. However, contrast analysis (n-BTC vs. n-ATC) revealed a strong activation in bilateral inferior frontal gyrus (IFG, triangular part) only when US was presented below the HT (p < 0.001, cluster > 30). In addition, IFG activation was also associated with faster reaction times during n-BTC (p = 0.033) as well as with verbal reports obtained after resting-state, i.e., the more unpleasant sound was perceived during BTC vs. ATC, the higher activation in bilateral IFG was and vice versa (p = 0.003). While this study provides no evidence for activation of primary auditory cortex in response to audible US (even though participants heard the sounds), it indicates that US can lead to changes in the cognitive control network and affect cognitive performance only when presented below the HT. Activation of bilateral IFG could reflect an increase in cognitive demand when focusing on task performance in the presence of slightly unpleasant and/or distracting US that may not be fully controllable by attentional mechanisms.

are imposed by the German Psychology Association (DGP). As all the data is being stored on password-protected internal servers of the Max Planck Institute for Human Development, any request for data could be send to schmalen@mpib-berlin.mpg.de.

**Funding:** Christian Koch has received funding from the EMPIR programme co-financed by the Participating States and from the European Union's Horizon 2020 research and innovation programme. The funders had no role in study design, data collection and analysis, decision to publish, or preparation of the manuscript. https://www.euramet.org/research-innovation/research-empir/ https://www.ffg.at/en/europe/h2020.

**Competing interests:** The authors have declared that no competing interests exist.

# Introduction

Technological progress and urbanization significantly contributed to the fact that nowadays air-conducted sound in the ultrasonic frequency range (> 17.8 kHz, US) represents an integral part of our daily stimulus environment. While the use of US in animal communication, medical technology and the manufacturing industries is sufficiently well-known, less attention is paid to the fact that US is also emitted by an increasing number of commercially available devices, which often results in people being exposed to such frequencies on a daily basis without noticing. Acoustic field measurements revealed that significant levels of US are present in a variety of public places such as railway stations, libraries, or schoolrooms [1–4]. Currently, two types of sources are considered the largest contributors of US in the public domain: Public address voice alarm (PAVA) systems, often found in shopping centers, airports, or football stadiums, capable of generating 20 kHz tones of up to 80 dB [2, 5, 6] as well as ultrasonic pest repellents, emitting US of over 100 dB [2, 7, 8]. Interestingly, there is still a widespread belief among the public that US refers to sound above 20 kHz and categorically exceeds the human hearing range. However, in recent years, there has been a growing consensus among researchers to define 17.8 kHz (i.e., the lower limit of the third octave band centered at 20 kHz) as the lower limit of the ultrasonic frequency spectrum [9]. It has been shown repeatedly that at a sufficiently high sound pressure level (SPL), hearing thresholds (HT) for air-conducted US at frequencies of up to 28 kHz can be determined [10–12]. Importantly, these studies also demonstrated that even in demographically homogenous samples, HTs for US stimuli vary widely from person to person, which suggests that certain individuals may be significantly more susceptible to the effects of US exposure than others.

To date, adverse effects of ultrasonic noise on workers in the manufacturing industries have been described in numerous studies and reports dating back to the 1940s, including a wide range of auditory (e.g., HT shifts or tinnitus) as well as non-auditory symptoms (e.g., dizziness, fatigue, nausea or migraine) [review in 13]. Meanwhile, a number of recent studies also addressed the question, whether similar symptoms could arise from US exposure in the public domain. Fletcher et al. [14] exposed participants to audible sound at frequencies between 13.5 kHz and 20 kHz and showed that these sounds were perceived as significantly more 'unpleasant' compared to a 1 kHz control stimulus. Interestingly, a subset of participants who had previously complained about alleged adverse effects of public US exposure also reported greater 'difficulty concentrating' and greater levels of 'annoyance'. Ueda et al. [7] investigated the effects of audible US by installing rodent repellents, capable of producing sound with a spectral peak at around 20 kHz and SPLs between 90 dB and 130 dB outside of a public restaurant and reported that all 35 participants aged 20 years to 50 years were able to hear the sound, with about half of them experiencing it as 'uncomfortable', 'noisy' and even causing 'pain in the ear'. Taken together, these studies seem to confirm the suspicion that once US reaches the HT, it immediately causes discomfort [1, 12] and–depending on other factors such as sound intensity, duration of exposure, or individual susceptibility–may also produce other adverse effects. In contrast, little is known about whether inaudible US could also have harmful effects. To address this question, Fletcher et al. [15] conducted a double-blind follow-up study, in which participants were exposed to an inaudible 20 kHz tone for 15 min vs. sham. Interestingly, while stimulation per se did not cause any adverse symptoms, the authors reported that (false) expectations about the presence of US led to small nocebo effects (i.e., slightly higher ratings for 'ear pain', 'dizziness', and 'tinnitus'). In a recent fMRI-study, Ascone et al. [16] also investigated potential adverse effects of inaudible US vs. sham by installing commercially available US sources capable of producing sound at about 22.4 kHz in participants' bedrooms. The authors reported that prolonged US exposure over a period of 28 consecutive nights had no

influence on any of the behavioral domains studied (such as sound sensitivity, quality of sleep or cognitive performance), while sham stimulation was again associated with minor nocebo-effects (i.e., increased somatization and phasic alertness). However, they also showed that US exposure was associated with significant reductions in grey matter volume in brain areas involved in executive functions, such as attention control and inhibition. While both studies stress the importance of individual expectations and beliefs in assessing US-related health effects, Ascone et al.'s findings indicate that inaudible US could also exert an influence on the central nervous system (CNS) in the absence of auditory perception, as has been suggested for sound with frequencies below 20 Hz (i.e., infrasound) at levels near the HT [17].

Although hearing in the US frequency spectrum appears to be restricted significantly by the poor impedance match of the middle ear at high frequencies [18, 19] and potential frequency limitations due to the tonotopic architecture of the cochlea [20], it most likely involves processing at the base of the cochlea (i.e., the high-frequency part) and is, therefore, an auditory sensation [3]. However, given that US in the public domain is often inaudible, Leighton [1] put forward an alternative hypothesis that could explain how some of the non-auditory symptoms attributed to US could emerge in the absence of auditory processing, i.e., via activation of mechanical proprioceptors in the tympanic membrane as well as muscles of the eustachian tube. The resulting discrepancy between proprioceptive input and inner ear activity could thus lead to symptoms such as headache, dizziness, nausea or fatigue. While this hypothesis has not been tested explicitly, Job et al. [21] showed that applying small air pressure changes to the tympanic membrane causes changes in brain activity at the caudal edge of the somatosensory cortex, which may be a first indication that inaudible US could affect signal processing in the somatosensory pathway.

To date, several studies have aimed to identify the neural correlates of US processing in humans by measuring brain activity in vivo during stimulus application, with overall mixed results. Using magnetoencephalography (MEG), Hosoi et al. [22] demonstrated for the first time that bone-conducted US with frequencies of up to 40 kHz led to event-related potentials in primary auditory cortex (PAC) approximately 100 ms after stimulus onset. Fujioka et al. [23] then investigated the effects of air-conducted US by means of MEG, yet no changes in brain activity were observed when stimulation exceeded 14 kHz, even though some participants reported a hearing impression at up to 20 kHz. However, it has been demonstrated in subsequent studies that the HT for a 20 kHz tone is usually higher than 85 dB SPL [overview in 12]. This indicates that stimulation may have been too weak to reliably elicit a hearing impression and/or brain effects, since stimuli were applied via a loudspeaker 135 cm in front of the participants at only 60 dB SPL. Nittono [24] conducted an EEG study, in which the cortical response towards high-frequency components of high-resolution audio was compared to similar sounds without these components, yet no differences were detected. Again, since the participants' mean HT was only 17,316 Hz and the test stimuli (a 11- and a 22-kHz high-cut sound) were delivered via loudspeakers 120 cm in front of the participants with an SPL of 62 dB, the 22-kHz stimulus may have been too weak to produce measurable effects. However, when trying to characterize the neural response to air-conducted US by means of MEG and fMRI, with stimuli being delivered via an in-ear-device 5 dB above the individual HT [12], the authors also found no evidence for auditory processing at frequencies above 14 kHz. Given the fact that the experimental conditions in the above-mentioned studies differed drastically, limiting overall data comparability, it would be premature to argue that MEG and fMRI are insufficient threshold detectors for air-conducted sounds in the US frequency range. Nevertheless, it remains unclear whether the lack of evidence for auditory processing can be attributed to limited sensitivity or low signal-to-noise ratio of the measurement tools, or whether other experimental conditions had a significant influence.

This study represents a follow-up to Kühler et al.'s [12] experiments using a modified experimental setup, while also addressing slightly different research objectives, i.e., investigating the effects of US on brain activity during resting-state as well as during cognitive processing. In the study, fMRI was used to investigate the neural response to air-conducted US presented above as well as below the individual HT in 15 normal-hearing individuals. In the first experiment, US stimuli were presented during 'resting-state', i.e., participants were asked to lie in the scanner calmly with eyes closed, not thinking of anything in particular and verbal reports regarding their hearing impression were obtained after each run. During resting-state, a characteristic pattern of large-scale brain activity emerges, which commonly involves the activation of several brain regions, such as medial prefrontal cortex (MPFC), posterior cingulate cortex (PCC), inferior parietal lobe (IPL), lateral temporal cortex (LTC), and the hippocampal formation (HC) [25, 26]. As demonstrated in a previous study, in which the neural response to infrasound was investigated [17], resting-state fMRI proved to be well suited for examining the effects of stimulation under conditions more similar to those in our daily environment, i.e., when humans are exposed to sound over a prolonged period of time. Moreover, since some of the statistical variance of resting-state fMRI data can be explained by the heterogeneity of participants' mental states during image acquisition [27, 28], the combined use of resting-state fMRI and verbal reports allows us to enter a mutually informative discourse between findings on the neural and on the perceptual level to best characterize the effects of stimulation. In general, we expected US to be experienced as at least slightly unpleasant and cause disruption of resting-state brain activity, most likely accompanied by PAC activation in response to US presented above the HT. In the second experiment we asked whether air-conducted US also affects cognitive processing. We expected that US would exert a negative influence on task performance in a visuo-spatial working memory task (n-back), accompanied by functional activity changes in the cognitive control network. We also expected participants, scoring higher on rating scales for depression, anxiety or neuroticism, to be more prone to experience detrimental performance and/or brain activation effects.

## Experimental procedures

### 1. Participants

21 healthy participants took part in the study based on written informed consent. 6 participants were excluded after the hearing threshold assessment and the remaining 15 participants underwent data acquisition by means of fMRI (8 female, mean age = 26.5, SD = 3.42; 8 male, mean age = 24.43, SD = 2.76). The study was conducted according to the Declaration of Helsinki (64th WMA General Assembly, 2013) with approval of the ethics committee of the German Psychological Association (DGPs). All 15 participants had normal or corrected-to-normal vision and were able to perceive test stimuli up to a frequency of at least 21.5 kHz. No participant had a history of neurological, major medical, or psychiatric disorder. All participants were right-handed as assessed by the Edinburgh handedness questionnaire [29].

### 2. Ultrasound source & sound system

To present pure sine tones in the US frequency spectrum during the HT assessment as well as during neuroimaging, a special sound source was developed in-house (PTB). This device was built to meet the requirements of the scanner environment by reducing the amount of metal, thus minimizing electromagnetic interference during data acquisition and was capable of producing US at frequencies of up to 40 kHz at 130 dB SPL (see Fig 1A for a photograph of the sound source and Fig 1C for a schematic drawing of the sound system). No artefacts due to electromagnetic interference were detected in the MRI measurements with the sound source

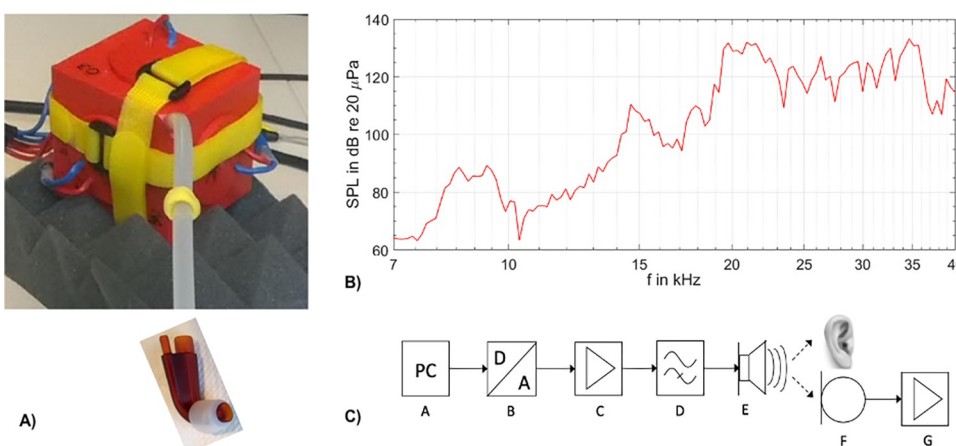

**Fig 1. Stimulus setup.** A) Photograph of the sound source with the proximal part of the silicone tube leaving the source at the upper corner of the cube and the distal part being attached to an adapter with a silicone ear plugs at its distal end. B) Frequency response of the sound source: Due to the spatial arrangement of the piezoelectric transducers the sound source exhibited several resonance peaks (i.e., 14.6 kHz, 17.9 kHz, 18.9 kHz, 21.5 kHz, 22.5 kHz and 26.4 kHz) which were utilized for stimulation. C) Schematic drawing of the sound system: A: Personal computer, B: D /A converter, C: Power amplifier, D: High pass filter, E: Sound source, F: Optical microphone, G: Microphone amplifier.

positioned just outside of the head coil. Sound signals were generated by a personal computer (see Fig 1A and 1C) and fed through a D / A converter (B, RME FireFace UC sound card), a power amplifier (C, t-amp Proline 1800) and a high-pass filter (D, built in-house) with a corner frequency of 18 kHz and a roll-off of 36 dB/octave before entering the scanner room. The purpose of the high-pass filter was to suppress technical noise and to protect participants from dangerously high SPLs in the audible range. The sound signals were then relayed to the sound source (E), which was attached to a silicone tube (length = 30 cm, inner diameter = 6 mm). The proximal part of the silicone tube was inserted at a corner of the cube leading to the cube center and the distal part led to the participants' right ear. The sound source (E) consisted of a cube made out of polyvinyl chloride (PVC; 'Trovidur') with an edge length of about 11 cm. Six piezoelectric transducers (Kemo L010) were fitted to the inside of the cube, one on each inner surface. The transducers were driven in phase to generate pure tones with sufficiently high SPLs for auditory perception. The proximal part of the silicone tube was inserted at a corner of the cube leading to its center. Due to the spatial arrangement of the piezoelectric transducers, the transfer function of the sound system was not smooth, but instead produced several resonance peaks. Fig 1B shows the maximum SPL achieved by the sound system when run with 250 mV and a sound card attenuation of 29 dB. The sound source did not produce any subharmonic distortions, however, harmonic distortions were clearly visible with the second harmonic about 30 dB, and the third harmonic about 40 dB lower than the first harmonic. As HTs are not expected to decrease at frequencies above 26.4 kHz, harmonic distortions were most likely not recognized by the participants.

The source was encased in foam, placed inside a plastic box and mounted to a wooden board next to the head coil via a hook and loop fastener. As participants were moved into the scanner, the sound source moved along with them, thus minimizing the risk of earplug displacement due to pulling forces exerted during positioning and allowing for sound generation in close vicinity to the participants' head. A custom-made ear adapter mounted to the distal portion of the silicone tube was used to deliver US directly into the participant's ear canal and also to monitor SPLs of the presented stimuli. For each participant, one of three adapters (inclination 75˚, 90˚, and 105˚) was selected to optimally match the anatomy of the

participants' ear and a silicone plug was connected to the distal portion of the adapter to ensure a firm yet comfortable fit (see Fig 1A). The pick-up opening was connected to an optical microphone via a second silicone tube (length = 30 cm, inner diameter = 2 mm) (F, Sennheiser Mo2000) and the microphone signal was relayed to a measuring amplifier (G, Brüel & Kjaer 2636), which allowed for in situ monitoring of SPLs during stimulus presentation.

For SPL calibration, the pick-up opening was connected to a ¼ in. measuring microphone (Brüel & Kjaer 4136) placed in volume approximating that of the ear canal (volume = 1 cm$^3$, estimated for the space between the tip of ear adapter and the tympanic membrane) [also see 30]. On its distal end, a foam earplug (ER3-14A) was inserted to achieve optimal sealing of the volume. The sound characteristics of the experimental setup were preserved by the following modifications to the foam earplug: the sound delivery tube was substituted by a tube with a larger diameter, equal to that of the ear adapter. A second tube was inserted, connecting the optical microphone. For sound administration, the silicone plug was insertion into the ear canal and an 8 kHz test tone (2550 ms on, 250 ms off, SPL less than 10 dB above the individual HT) was presented, which helped finding the optimal position for adapter placement until participants reported optimal fit as well as a loud and clear hearing impression. Once positioned, the adapter was secured by two strips of tape placed over the auricle. Both earplug and adapter could be worn comfortably underneath additional ear protection. A regular earplug (E-A-R One Touch, 3M, St. Paul, USA) with a Noise Reduction Rating (NRR) of 33 dB was used for the left ear and both ears were covered with a Silverline 140858 ear defender (NRR: 22 dB) to minimize the interference of scanner noise during data acquisition.

## 3. Hearing threshold assessment & stimulus characteristics

Prior to the fMRI scan, HTs for a number of sounds in the very-high as well as in the US frequency spectrum were assessed in all participants by means of a modified Békésy procedure [31], described below. To take part in the fMRI experiment, participants were required to reliably detect sounds at > 20 kHz with a maximum SPL of 125 dB (i.e., 130 dB safety limit minus 5 dB 'headroom' for the above-threshold condition). Out of 21 participants who underwent an initial screening process, 15 met these requirements and were thus eligible for further data acquisition. In five randomly chosen participants the HT assessment was repeated on two non-consecutive days to ensure data reproducibility. HTs for pure tones with frequencies ranging from 8 kHz to 26.5 kHz were measured monaurally (right ear) while participants lay in the scanner to ensure that US audibility was preserved during the imaging process (no scan was performed but the scanner's background noise level was above the limit according to ISO 8253–1 [32]). Stimulus frequencies for the HT assessment were chosen so that the resonance peaks of the sound source could be utilized (i.e., 14.6 kHz, 17.9 kHz, 18.9 kHz, 21.5 kHz, 22.5 kHz and 26.4 kHz).

The measurement was carried out according to a modified Békésy procedure [31] with three iterations, starting from the lowest frequency (see Fig 2 for a schematic of the hearing threshold assessment paradigm). Participants fully controlled the procedure via a button box placed in their right hand. Pushing the right button led to an increase in SPL by 2 dB and pushing the left button led to a decrease in SPL by 2 dB. 500 ms after each button was pressed a tone stimulus of 1000 ms duration was presented. Prior to this assessment, participants were informed that the tone may at first be inaudible and that several right button presses may be required for the tone to reach a SPL at which it can be perceived. To determine the HT, participants were asked to indicate the SPL at which a tone of a given frequency was barely audible (defined as 'one right button press above inaudibility') by pressing the middle button. Participants were also informed that precise identification may require several approaches to the HT

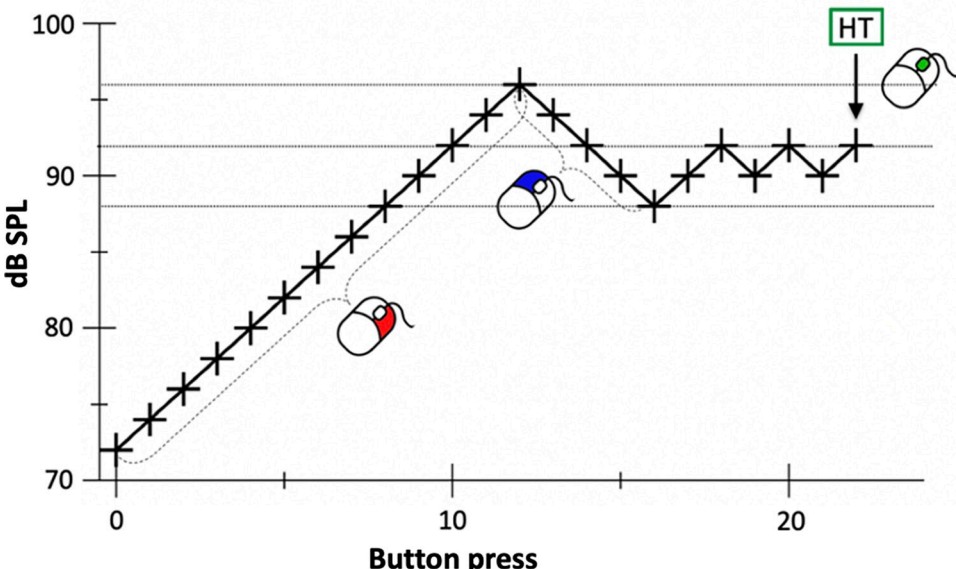

**Fig 2. Schematic of the hearing threshold assessment paradigm.** Exemplary data of one participant determining her hearing threshold (HT) for a given test frequency. The participant was asked to gradually increase the SPL of a given US stimulus in steps of 2 dB until the stimulus just became audible and then to level out at the definitive HT by approaching it repeatedly from SPLs above and below (i.e., pressing the right button increased the SPL by 2 dB and pressing the left button reduced the SPL by 2 dB). The correct SPL (i.e., 'one right button press above inaudibility') was registered via a middle button press and followed by presentation of the next stimulus.

from SPLs above and below. If the maximum SPL (pre-defined according to either the maximum technical output of the sound source or the ethical constraint of 130 dB) for a given stimulus had been reached, participants received visual feedback via a computer screen. On average, participants performed 20.5 button presses (SD = 9.5), until the HT for each frequency was identified. Since all 15 participants were able to detect US at 21.5 kHz across all three iterations, individual HTs for that particular frequency were used to define stimuli for the subsequent fMRI experiments. During imaging, US was presented in the form of tonebursts with a duration of 2550 ms, on- and offset ramps of 100 ms and an interstimulus interval of 250 ms either 5 dB above the individual HT ('above-threshold condition', (n-)ATC) or 10 dB below the individual HT ('below-threshold condition', (n-)BTC).

## 4. Scanning procedure

Images were collected on a 3T Verio MRI scanner system (Siemens Medical Systems, Erlangen, Germany) using a 12-channel head coil. First, high-resolution anatomical images were acquired using a three-dimensional T1-weighted magnetization prepared gradient-echo sequence (MPRAGE), repetition time = 2300 ms; echo time = 3.03 ms; flip angle = 9˚; $256 \times 256 \times 192$ matrix, $(1 \text{ mm})^3$ voxel size. Whole-brain functional images were collected using a T2*-weighted EPI sequence sensitive to BOLD contrast (TR = 2000 ms, TE = 30 ms, image matrix = $64 \times 64$, FOV = $(200 \text{ mm})^2$, flip angle = 80˚, slice thickness = 3.5 mm, 35 nearaxial slices, aligned with the AC/PC line). Before resting-state data acquisition started, participants had been in the scanner for about 10 minutes. During those 10 minutes, a localizer was run and structural images were acquired so that participants could get used to the scanner noise. To ensure that participants were exposed to a minimum of scanner-induced background noise, the cryo-cooler compression pump system was switched off for the entire duration of the fMRI scans.

**4.1. fMRI protocol–Resting state.** The fMRI protocol comprised a total of six sequences, three resting-state runs, followed by three n-back runs. During resting-state, each participant underwent one unstimulated and two stimulated runs, each lasting 300 s. During the unstimulated run, participants were scanned in the absence of auditory stimulation ('no-tone condition'; NTC), while during the two stimulated runs a 21.5 kHz US tone was presented either at 5 dB above ('above-threshold condition'; ATC) or 10 dB below the participants' HT ('below-threshold condition'; BTC). Before the start of each run, participants were instructed to keep their eyes closed, relax and not to think of anything in particular. Resting-state data acquisition always started with the NTC run, whereas the order of the stimulated runs was alternated across participants. Analysis of the resting-state data included 8 data sets in which ATC was followed by BTC, and 7 in which the order was reversed. The runs were conducted in a single blind fashion, i.e., only the experimenter knew the order of the stimulus conditions.

**4.2. fMRI protocol–N-back task.** Cognitive performance was assessed by means of a visuo-spatial working memory task (n-back), divided into three consecutive runs, each consisting of 12 blocks (see [33] for a similar experimental design to assess cognitive performance during infrasound exposure). In each block, a sequence of 10 black dots appeared at varying locations in a 4 by 4 grid. By using a button box placed in their right hand, participants were asked to indicate whether each dot appeared at the same position as the dot presented three steps earlier in the sequence ('3-back'). Pressing the left button signaled a match, whereas pressing the right button signaled a mismatch. Each black dot was presented for 3000 ms at a pseudo-randomized location in the grid with the constraint of not appearing at the same location in two consecutive steps. All 12 blocks differed in dot order to avoid memory effects. During n-ATC and n-BTC blocks, repeated bursts of US with a frequency of 21.5 kHz, a duration of 2550 ms, on- and offset ramps of 100 ms and an inter-stimulus interval (ISI) of 250 ms were administered in temporal alignment with dot presentation. In each run, the 12 blocks were aligned with the stimulus protocol in the following order: n-NTC, n-BTC, n-ATC, n-NTC, n-BTC, n-ATC, n-ATC, n-BTC, n-NTC, n-ATC, n-BTC, n-NTC (see Fig 3 for the stimulus paradigm and the behavioral task in a n-back block). This sequence was chosen with the aim of minimize the risk of stimulation order affecting task performance and/or brain activation (by alternating ATC and BTC) as well as the risk of data acquisition being influenced by whether a stimulation run was followed by a run without stimulation (by starting with NTC in the first 2 triplets and ending with NTC in the last 2 triplets) within a given amount of scanning time. After each block, a fixation period of 20 s was inserted, during which participants were looking at a black cross in the middle of the screen.

## 5. Questionnaires & verbal reports

Beck Depression Inventory (BDI-II) [34] was used for the assessment of depressive symptoms, the State-Trait Anxiety Inventory (STAIX1/X2) [35] to measure state- and trait-anxiety. Neuroticism was rated via the Big Five personality traits short-scale (BFI-S) [36]. After each of the n-back and resting-state runs, participants were asked a number of questions regarding their hearing impression. After each of the six runs, participants were asked 'Did you hear sounds during the last run?' and asked to answer with 'yes', 'no' or 'unsure', if participants could not reliably distinguish between scanner noise and stimulus. If participants answered 'yes' or 'unsure' after a resting-state run they were then asked 'On a scale between -5 (extremely unpleasant) and +5 (extremely pleasant), how did you perceive the sounds'? If the participants answer 'yes' after an n-back run, they were then asked 'On a scale between -5 (extremely negative) and +5 (extremely positive), how much did the sounds influence your performance?".

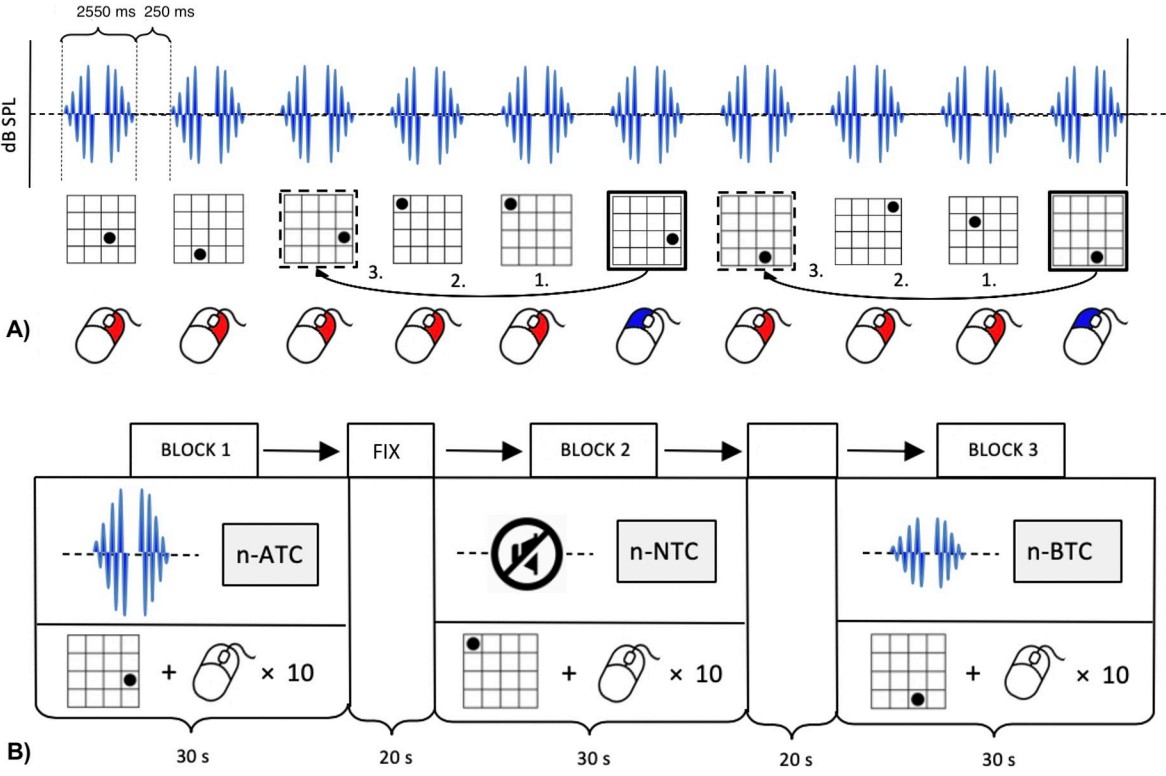

**Fig 3. Stimulus paradigm and schematic drawing of the n-back run.** A) One n-back block consisted of 10 brief bursts of US with a frequency of 21.5 kHz, a duration of 2550 ms and an inter-stimulus interval (ISI) of 250 ms presented either above (n-ATC) or below (n-BTC) the participants' HT (n-NTC not depicted here). Each tone presentation corresponded to a black dot presented for 3000 ms at a varying location on a four-by-four grid. Participants reported whether each dot appeared at the same position as the dot presented three steps earlier in the sequence by pressing one of two buttons (right = match; left = mismatch). B) One n-back run consisted of 12 blocks each lasting 30 s with each of the conditions (NTC, ATC, BTC) presented three times in random order. Between consecutive blocks, a fixation period of 20 s was inserted.

## 6. Resting state data analysis

The first three volumes of each run were discarded to allow the magnetization to approach a dynamic equilibrium. Part of the data pre-processing, including slice timing, head motion correction (a least squares approach and a 6-parameter spatial transformation) and spatial normalization to the Montreal Neurological Institute (MNI) template (resampling voxel size of 3 mm × 3 mm × 3 mm) were conducted using SPM12 and the Data Processing Assistant for (Resting-State) Brain Imaging (DPABI [37]). A spatial filter of 4 mm FWHM (full-width at half maximum) was used. Participants showing head motion above 3 mm of maximal translation (in any direction of x, y or z) and 1.0˚ of maximal rotation throughout the course of scanning would have been excluded. After pre-processing, linear trends were removed. Then the fMRI data was temporally band-pass filtered (0.01–0.08 Hz) to reduce low-frequency drift and high-frequency respiratory and cardiac noise [38]. Resting state data was approached using regional homogeneity (ReHo) as well as by a seed-based approach using functional connectivity analysis. ReHo analysis [39–42] was performed using DPABI. It is a technique that captures the synchrony of resting-state brain activity in neighboring voxels (i.e., local connectivity) and allows to track changes in activity anywhere in the brain without having to pre-define a region of interest (ROI) [43, 44]. ReHo was originally invented for the analysis of (slow) event-related fMRI data [39], but is equally suited for block-design and resting-state fMRI. For each

participant, ReHo analysis was performed on a voxel-wise basis by calculating the Kendall's coefficient of concordance (KKC [45]) of the time series of a given voxel with those of its neighbors (26 voxels). The KCC value was assigned to the respective voxel and individual KCC maps were obtained. ReHo was calculated within a brain-mask, which was obtained by removing the tissues outside the brain using the software MRIcro (http://people.cas.sc.edu/rorden/mricron/install.html)). Whole-brain comparisons between conditions were computed on the basis of the resulting ReHo maps. A height threshold of p < 0.001 and cluster-size corrected by means of Monte Carlo simulation (10000 iterations) was used. Significant effects were reported when the volume of the cluster was greater than the Monte Carlo simulation-determined minimum cluster size for the whole-brain volume, above which the probability of type I error was below 0.05 (AlphaSim [46]). Then mean ReHo values were extracted from the functionally defined PAC, based on the Anatomy Toolbox [47]. Functional connectivity from PAC as a seed was also computed using DPABI.

## 7. fMRI task-based data analysis

The fMRI data were analyzed using SPM12 software (Wellcome Department of Cognitive Neurology, London, UK). In addition to three scanner administered saturation volumes four more volumes of all EPI series were excluded from the analysis to allow the magnetization to reach a dynamic equilibrium. Data processing started with slice time correction and realignment of the EPI datasets. A mean image for all EPI volumes was created, to which individual volumes were spatially realigned by means of rigid body transformations. The structural image was co-registered with the mean image of the EPI series. Then the structural image was segmented and normalized to the Montreal Neurological Institute (MNI) template for the random-effects analysis. The normalization parameters were then applied to the EPI images to ensure an anatomically informed normalization. A commonly applied filter of 8 mm FWHM (full-width at half maximum) was used. Low-frequency drifts in the time domain were removed by modelling the time series for each voxel by a set of discrete cosine functions to which a cut-off of 128 s was applied. The statistical analyses were performed using the general linear model (GLM). We modelled conditions with above and below HT stimulation and without tone presentation as separate regressors. For the n-back analysis entire blocks were modelled. These vectors were convolved with a canonical hemodynamic response function (HRF) and its temporal derivatives to form regressors in a design matrix. Furthermore, six movement regressors were entered into the GLM. The parameters of the resulting general linear model were estimated and used to form contrasts. The resulting contrast image was then entered into one sample T-tests at the second (between-subject) level. Typical fMRI analyses include between 100000 to 200000 voxels resulting in numerous statistical tests, which must be appropriately corrected for multiple comparisons. Instead of testing each voxel individually, most fMRI analyses test whether a given cluster of voxels exhibits statistically significant activation, assuming that activations in proximate voxels are not fully independent. To display the results of the group analysis, statistical values were thresholded with a level of significance of p < 0.001 (z > 3.09, uncorrected); a significant effect was reported when the volume of the cluster was greater than the minimum cluster size determined by Monte Carlo simulation above which the probability of type I error was < 0.05.

## Results

### 1. Hearing threshold data

Average HTs for each frequency (i.e., 14.6 kHz, 17.9 kHz, 18.9 kHz, 21.5 kHz, 22.5 kHz and 26.4 kHz, with three iterations per frequency) measured via the optical microphone are

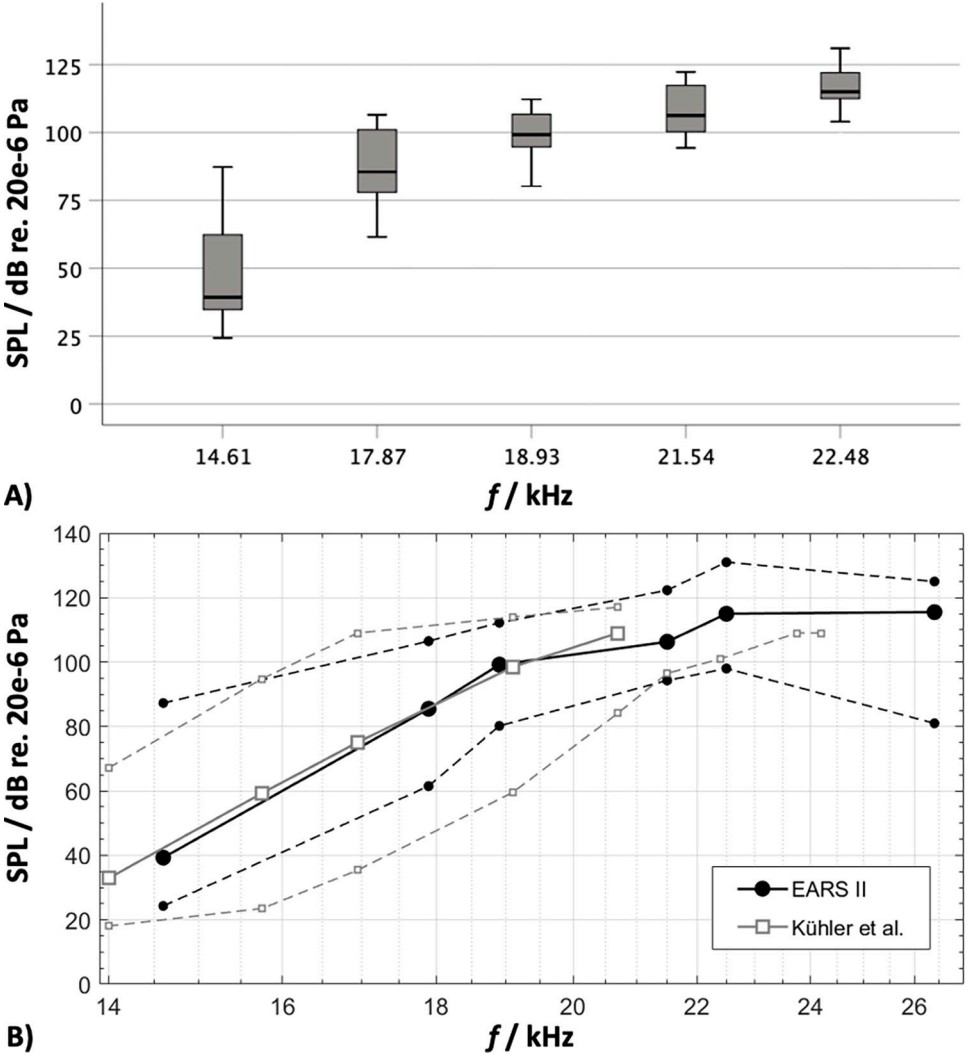

**Fig 4. Results of the hearing threshold assessment.** A) Hearing thresholds (HT) of 15 participants in response to monaurally presented pure tones at frequencies of 14.6 kHz, 17.9 kHz, 18.9 kHz, 21.5 kHz, 22.5 kHz and 26.4 kHz. Data is depicted as box-plots, showing medians (black lines), 25% and 75% percentiles as well as minimum and maximum values (whiskers) for each frequency. B) HT data from the present study (black) as well as Kühler et al. [12] (grey) depicted as median, minimum and maximum for each stimulus frequency.

depicted as box plots in Fig 4A. Whereas all 15 participants were able to determine their HTs for frequencies ranging from 14.6 kHz up to 22.5 kHz, only 8 participants reportedly heard a sound at 26.4 kHz. As can be derived from Fig 4A, HTs increased markedly from 14.6 kHz to 18.9 kHz, followed by a more asymptotic course at 22.5 kHz and 26.4 kHz. HTs varied significantly with respect to stimulus frequency, ranging from a median SPL of 39.3 dB at 14.6 kHz to 115.5 dB at 26.4 kHz (always given as dB re 20 μPa reference sound pressure), and as reported in previous studies (e.g. [12]), the range between the minimum and maximum HT for each frequency also varied significantly between participants. Interestingly, the largest variability was observed when sounds at 14.6 kHz were presented (range = 63 dB, SD = 19.0 dB). Variability then tended to decrease as frequency increased to 21.5 kHz (range = 28 dB, SD = 9.7 dB) and at 26.4 kHz the inter-individual spread increased again (range = 44 dB, SD = 14.3 dB). Overall, median values of the HTs were in good agreement with previous

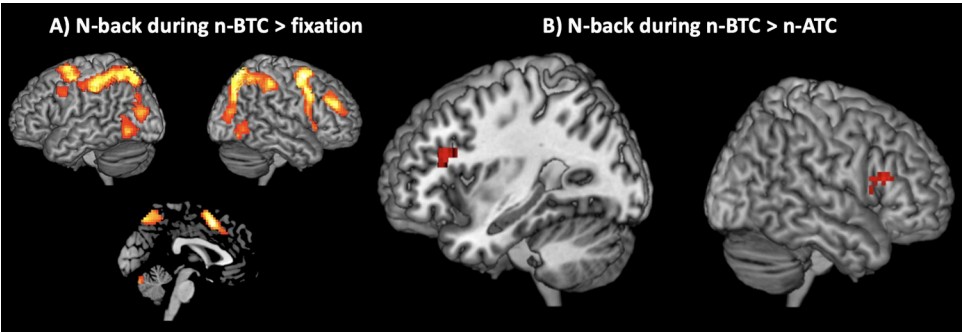

**Fig 5. Results of whole-brain contrast maps acquired during n-back.** A) N-back during the no-tone condition (n-NTC) vs. fixation showed significant activation within prefrontal and parietal cortex, as well as in the cerebellum, indicating the recruitment of a cognitive control network (p < 0.001, cluster > 20). B) Comparing data gathered during n-back 10 dB below (n-BTC) vs. 10 dB above (n-ATC) the individual hearing threshold (HT) showed pronounced activation in bilateral inferior frontal gyrus (IFG; triangular part) only in the n-BTC condition (peak voxels according to MNI: -36, 23, 22 for left and 48, 23, 16 for right IFG) (p < 0.001, cluster > 30).

studies [10–12] (see Fig 4B). Since the 21.5 kHz tone was detected reliably by all 15 participants with a HT at least 5 dB below the stimulation maximum of 130 dB SPL, this tone was selected for the subsequent fMRI investigation. The median monaural HT for a 21.5 kHz pure tone was 106.3 dB SPL, ranging inter-individually from 94.3 dB SPL to 122.3 dB SPL (SD = 9.7 dB). In five randomly chosen participants the entire HT assessment was repeated on two non-consecutive days to test and ensure data reproducibility. Remarkably, intra-individual scattering only varied between -1 and +5 dB SPL.

## 2. fMRI task-based data

To assess whether n-back task performance was associated with the recruitment of brain areas known to be involved in working memory, we contrasted whole brain data gathered during n-NTC vs. fixation in a block analysis and found activation of the typical cognitive control network, comprising prefrontal, parietal and cerebellar brain areas (Fig 5A, Table 1). Next, we

**Table 1. N-back without tone (n-NTC) vs. fixation.**

| Area | BA | Peak coordinates (MNI) | t-Score | Extent | pFDR |
|---|---|---|---|---|---|
| Left superior parietal lobe | 7 | -24, -61, 52 | 14.10 | 2097 | 0.000 |
| Anterior cingulate cortex | 32 | -3, 11, 49 | 12.18 | 1324 | 0.000 |
| Right cerebellum, Vermis | | 6, -61, 32 | 8.35 | 29 | 0.098 |
| Right dorsolateral prefrontal gyrus | 46 | 33, 41, 34 | 8.30 | 117 | 0.004 |
| Right inferior frontal gyrus | 44 | 54, 11, 7 | 8.23 | 146 | 0.001 |
| Left caudate | | -12, -4, 10 | 7.71 | 48 | 0.037 |
| Right cerebellum, Crus 1 | | 42, -58, -26 | 7.27 | 48 | 0.008 |
| Left cerebellum | | -39, -67, -26 | 7.16 | 104 | 0.005 |
| Left anterior insula | 13 | -30, 23, 10 | 6.32 | 54 | 0.030 |
| Right thalamus | | 12, -13, 1 | 5.86 | 24 | 0.118 |
| Left middle occipital gyrus | | -51, -73, 4 | 5.72 | 91 | 0.007 |
| Right pallidum | 19 | 15, -1, 13 | 5.37 | 39 | 0.057 |
| Right middle temporal gyrus | | 54, -61, 1 | 5.02 | 58 | 0.027 |
| Left cerebellum, Vermis | 37 | -9, -72, -26 | 4.96 | 25 | 0.118 |

(p < 0.001, k > 22). k, cluster size; BA, Brodmann areas; MNI, Montreal Neurological Institute; pFDR, positive false discovery rate.

addressed the question, whether stimulation during n-ATC or n-BTC led to significant differences in brain activation anywhere in the brain. To do so, we calculated pairwise contrasts between each stimulus condition and the no-tone condition separately, i.e., n-ATC vs. n-NTC and n-BTC vs. n-NTC. However no significant differences were found (thresholded at $p < 0.001$, uncorrected). Since we hypothesized that if air-conducted US can be processed by the CNS, it will most likely involve activation in PAC, we compared the same conditions again by using PAC as the ROI, yet no significant differences were found, even when restricting the analysis to only those participants who reported a hearing impression in the respective condition. However, when contrasting the two stimulation runs directly (n-BTC vs. n-ATC) we found a strong activation in bilateral inferior frontal gyrus (IFG, triangular part) only when US was presented below the HT ($p < 0.001$, cluster $> 30$) (peak voxels according to MNI: -36, 23, 22 for left and 48, 23, 16 for right IFG) (Fig 5B) as well as a trend towards higher activation during n-BTC compared to n-NTC ($p = 0.068$).

### 3. fMRI resting state data

No differences were observed when ReHo was computed on data gathered during the three resting-state acquisitions. The same holds true for functional brain connectivity at rest with PAC as a seed voxel.

### 4. Verbal reports & behavioral data

In a first step, we analyzed the verbal reports given by participants after each resting-state run. When participants were asked 'Did you hear sounds during the last run?' after NTC, 11 out of 15 accurately stated that no stimulation had taken place, 3 were unsure and 1 reported that sounds had been presented. In contrast, 10 out 15 participants reported that they heard sounds during ATC, but also 10 participants had a hearing impression during BTC, with one participant answering 'unsure' after each of the stimulus conditions. Those participants who reported that stimulus application had taken place or were unsure were then asked 'On a scale between -5 (extremely unpleasant) and +5 (extremely pleasant), how did you perceive the sounds?'. Here, paired t-test (two-tailed) revealed that stimulation during ATC (mean = -1.30, SD = 1.160) was rated significantly more unpleasant compared to BTC (mean = 0.20, SD = 0.789) (t(9) = -3.545, p = 0.006). In a second step, we analyzed the verbal reports after each n-back run. When participants were asked 'Did you hear sounds during the last run?' after the first n-back run, 7 out of 15 (one 'unsure'), after the second n-back run 10 out of 15, (one 'unsure'), and after the third n-back run 11 out of 15 participants reported that they perceived a sound. Those participants who answered 'yes' were then asked 'On a scale between -5 (extremely negative) and +5 (extremely positive), how much did the sounds influence your performance?'. During the first run, stimulation affected participants very differently, but there was no trend towards a perceived positive or negative influence on performance (mean = 0.00, SD = 1.51, min = -4, max = +3). In contrast, analysis of the verbal reports obtained after the second and third run showed that on average participants reported a slightly negative influence on task performance (mean for run 2 = -0.87, SD = 1.30, mean for run 3 = -0.80, SD = 1.42) (see Table 2). In a third step, we looked for correlations between these verbal reports and brain data gathered during resting-state as well as n-back performance. While no correlations between resting-state brain data and verbal reports reached significance, we found that bilateral IFG activation during n-BTC vs. n-ATC was associated with verbal reports after resting-state in two ways: First, we found that the more pleasant a tone was perceived during ATC, the higher activation in bilateral IFG during n-BTC vs. n-ATC was (r(14) = 0.579, p = 0.024). Second, this association remained significant when verbal reports during ATC

**Table 2. Verbal reports obtained after resting state as well as n-back performance.**

|  | Q1 | | | Q2 | | | | Q3 | | | |
|---|---|---|---|---|---|---|---|---|---|---|---|
|  | No | Yes | Unsure | Mean | SD | Min | Max | Mean | SD | Min | Max |
| NTC | 11 | 1 | 3 | 0.53 | 1.06 | -2 | 3 |  |  |  |  |
| BTC | 4 | 10 | 1 | 0.20 | 0.79 | -1 | 2 |  |  |  |  |
| ATC | 4 | 10 | 1 | -1.30 | 1.16 | -3 | 1 |  |  |  |  |
| N-Back run 1 | 7 | 7 | 1 |  |  |  |  | 0,00 | 1.51 | -4 | 3 |
| N-Back run 2 | 4 | 10 | 1 |  |  |  |  | -0,87 | 1.30 | -4 | 0 |
| N-Back run 3 | 4 | 11 | 0 |  |  |  |  | -0.80 | 1.42 | -4 | 1 |

Data was obtained after one unstimulated (NTC) and two stimulated resting-state runs ('below-threshold condition', BTC; 'above-threshold condition', ATC) as well as three n-back runs with all stimulus conditions presented according to a predefined sequence. Q1: 'Did you hear sounds during the last run?'. Q2: 'If yes, on a scale between -5 (extremely unpleasant) and +5 (extremely pleasant), how did you perceive the sounds'. Q3: "If yes, on a scale between -5 (extremely negative) and +5 (extremely positive), how much did the sounds influence your performance?".

were contrasted with BTC, i.e., the more unpleasant sound was perceived during BTC relative to ATC, the higher the activation in bilateral IFG was and vice versa (r(14) = 0.711, p = 0.003) (Fig 6A). In a fourth step, we looked for association between brain data and performance measures gathered during n-back. Here, we found that higher IFG activation was associated with faster reaction times (RTs) during n-BTC (r = -0.561, p = 0.033) (Fig 6B). In addition, the more unpleasant sound was perceived during BTC, the lower error rates during n-BTC were (r(14) = 0.633, p = 0.049) and n-ATC were (r(14) = 0.810, p = 0.005).

## 5. Data on personality factors

To test whether participants scoring higher on rating scales for depression, anxiety, and neuroticism would be more prone to experience detrimental performance or brain activation effects, difference scores between performance measures in n-back and brain activation scores with and without US exposure were calculated. We then correlated these scores with the sum scores of the Beck Depression Inventory (BDI-II), the State-Trait Anxiety Inventory (STAIX1/X2), and the Big Five personality traits short-scale (BFI-S), yet no significant correlations were observed.

## Discussion

The findings of the present study can be summed up in the following way: Contrary to our expectation, air-conducted US did not elicit activation in PAC in any of the experimental conditions (i.e., three resting-state runs and three n-back runs), irrespective of whether sound was administered below or above the HT and even though participants heard the sounds (as determined by HT assessments prior to as well as verbal reports obtained after the fMRI runs). However, contrast analysis of data gathered during n-back performance (n-BTC vs. n-ATC) revealed significantly higher activation in bilateral IFG only when US was presented below the HT. In addition, we found that the strength of this activation correlated with verbal reports obtained after the resting-state runs as well as with performance measures gathered during the n-back runs. First, we showed that the more unpleasant sound was perceived during BTC compared to ATC the higher signal strength in bilateral IFG was when comparing n-BTC to n-ATC (although ATC was generally experienced as more uncomfortable). Second, while RTs and error-rates for each condition did not differ significantly, higher signal strength in bilateral IFG was associated with faster RTs during n-BTC and third, error-rates during n-BTC and n-ATC were lower the more unpleasant sound during BTC was perceived. Referencing evidence

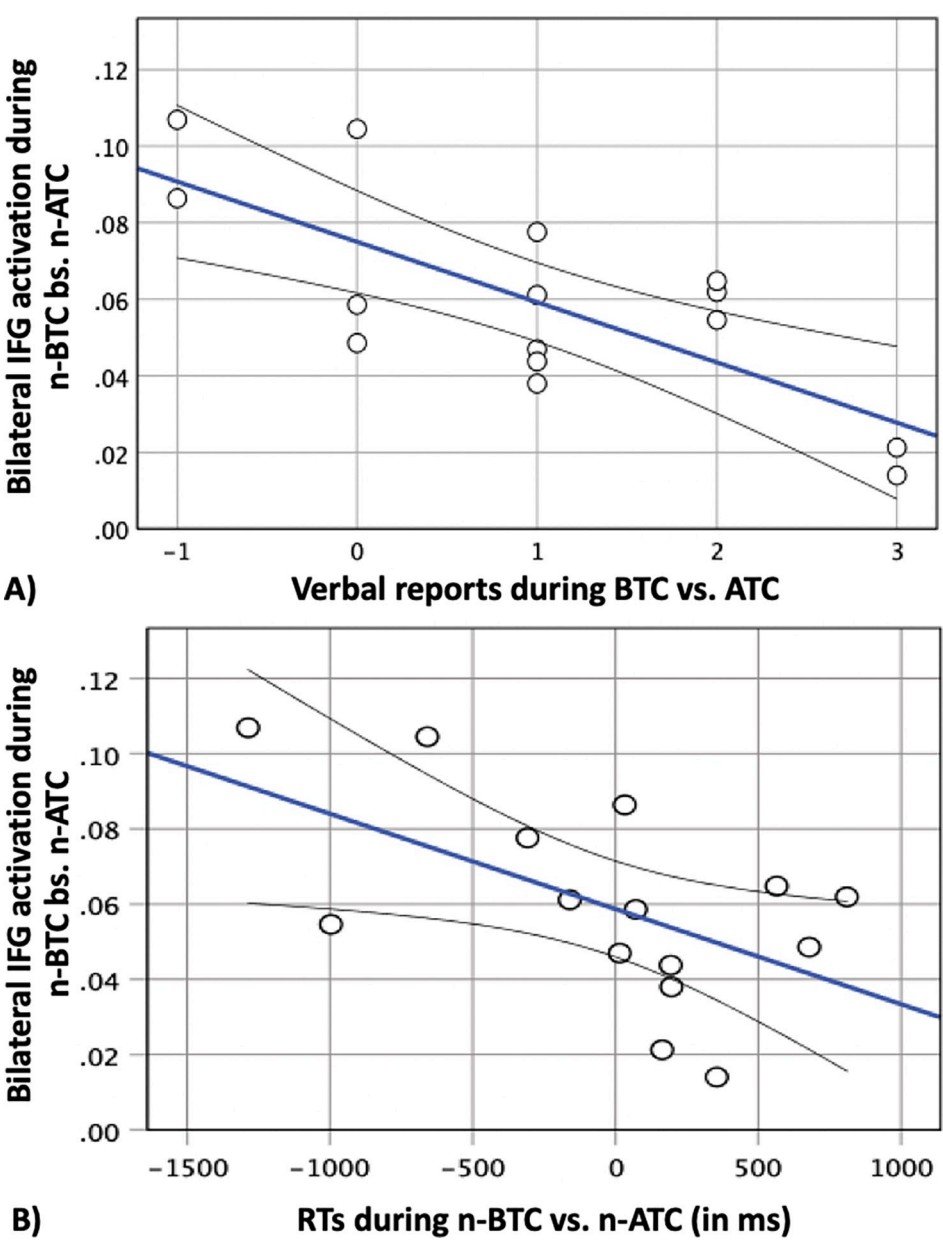

**Fig 6. Correlations between perceptual and n-back performance data with brain data gathered during task performance.** (A) The higher activation in bilateral inferior frontal gyrus (IFG) during n-back with stimulation below vs. above the hearing threshold (n-BTC vs. n-ACT) was, the more unpleasant sound was perceived during resting state with sound below vs. above the hearing threshold (BTC vs. ATC) (r = -0.711, p = 0.003) and (B), the faster reaction times (RTs) in the same contrast were (r = -0.561, p = 0.03).

for the involvement of IFG in several cognitive control processes, the authors argue that activation of bilateral IFG may reflect an increase in cognitive demand when focusing on task performance in the presence of unpleasant and/or distractive US. Yet it appears that higher IFG activation also helped those participants who were affected more negatively by US below the HT to maintain their level of cognitive performance despite interfering influences.

Working memory (WM) is commonly conceptualized as a system involving both storage and control processes that maintain access to information in the service of complex cognitive

activities [48]. Meanwhile, the n-back task has become one of the most popular tools for assessing WM under fMRI conditions [49, 50] and a number of studies confirmed that n-back performance critically relies on both 'traditional' WM-related functions (such as information storage and updating) as well as cognitive control processes (such as selective attention, inhibition and interference control) [51–55]. In addition, it is becoming increasingly clear that many of these cognitive functions share a common neural basis, which motivated the description of the so-called 'cognitive control network', linking WM and cognitive control in a coherent framework of functional brain activity [56–58]. As shown by means of fMRI, n-back performance commonly leads to activation of several key structures implicated in the cognitive control network, such as dorsolateral prefrontal cortex (DLPFC), premotor cortex (PMC), cingulate cortex (CC), posterior parietal cortex (PPC) and the cerebellum [57, 59]. The fact that there is a large overlap between these areas and the ones identified in our initial analysis (n-NTC vs. fixation) strongly suggests that instead of coincidental activation in task-irrelevant brain systems, the signal change detected in the latter part of the analysis (i.e., increased bilateral IFG activation during n-BTC vs. n-ATC) indeed reflects a change in cognitive processing attributable to changes in the stimulus environment. However, the question remains how to interpret this finding, given the complex involvement of IFG in a wide range of cognitive processes.

There is a growing amount of evidence that the functional organization of lateral PFC (Brodmann Area (8, 9, 10, 45, 46, and 47) follows a dorsal-ventral gradient with superior frontal cortex being more strongly involved in monitoring and manipulation of information at the service of executive processes, whereas the ventral frontal cortex is important for rehearsal during simple storage [60–63]. In addition, previous meta-analyses also provided evidence for a hemispherical separation. Here, right PFC appears to be more strongly involved in spatial WM, whereas left PFC is activated more strongly during verbal WM [59, 63, 64]. In particular, IFG (containing ventrolateral PFC) was found to be implicated in WM in a number of ways: For example, several studies showed that left IFG (BA 44 and 45—frequently referred to as Broca's area, the brain structure linked to speech production -, as well as BA 46) is critically involved in subvocal verbal rehearsal [65–67] and a meta-analysis conducted by Rottschy et al. [68] revealed that WM load is mainly associated with activation in bilateral IFG. Following this line of evidence, it appears that in contrast to other studies, in which WM load was modulated by increasing the number of items to be remembered, in the present account, cognitive demand may have increased depending on perceived discomfort and/or distraction during stimulation. This idea is supported by the fact that bilateral IFG activation was stronger the more unpleasant US during BTC compared to ATC was perceived and that overall, US was perceived as having a 'slightly negative' influence on task performance. In line with this notion, activation of left IFG could also indicate a stronger employment of metacognitive strategies such as subvocal verbal rehearsal to cope with increased cognitive demand.

Apart from that, there is ample evidence for the involvement of right IFG in other cognitive control processes. One line of research identified right IFG as a key component of the so-called 'ventral attention system', which was found to mediate the stimulus-driven aspects of attention, including attentional shifts to salient stimulus features [69–71]. Other studies showed that right IFG is also critically involved in auditory change detection, a process by which the CNS automatically identifies alterations in the auditory environment, including both simple changes (such as pitch or loudness) as well as more complex changes (such as grammar violations in mother-tongue sentences). Source reconstruction across different modalities confirmed that two processes contribute to auditory change detection, a bilateral supratemporal component as well as a (predominantly right-hemispherical) frontal component [review in 72]. Importantly, several authors have argued that the frontal component is not only involved

in the detection of deviant auditory stimuli, but also in the initiation of involuntary attention shifts in response to stimulus detection [73, 74], with right IFG being particularly important in these processes [75, 76]. In view of these findings, it is conceivable that US during n-BTC may have prompted participants to allocate attention away from task performance, thus increasing cognitive demand via distraction. Moreover, when analyzing the verbal reports obtained after the n-back runs, it becomes evident that as the number of participants who reported hearing US during n-back increased from run 1 (7 out of 15) to run 3 (11 out of 15), the perceived influence of stimulation on task performance also changed from neutral in run 1 (mean = 0,00) to slightly negative (mean = 0,80) in run 3. One could thus speculate that as participants became more experienced with task performance, they also became more susceptible to attentional shifts induced by US (and thus aware of stimulation and its effect). As a consequence, one would expect those participants who were distracted by US to a greater extent to also exhibit a deterioration of performance in the n-back task. However, the opposite was the case, as those participants with higher bilateral IFG activation showed significantly faster RTs during n-BTC compared to n-ATC and error-rates were also lower the more unpleasant sound during BTC was perceived. Taken together, these results rather suggest the presence of a compensatory effect, in the sense that higher IFG activation may have helped those participants who were affected more negatively by US below the HT to maintain their level of cognitive performance. This is particularly noteworthy, since these results seem to point in a similar direction as the trend towards WM improvement during infrasound exposure, reported in Weichenberger et al. [33]. In what way IFG may contribute to this effect remains unclear. However, it is worth mentioning that apart from the detection of salient stimulus features, IFG also supports a number of other cognitive control demands, such as sustained attention and inhibition of motor commands and intrusive thoughts [77–81]. One could thus speculate that inhibitory functions may have also played a role in supporting goal-directed behavior in a distractive environment by allowing attention to shift away from previously attended auditory stimuli.

Given the lack of evidence for primary sensory processing anywhere in the brain, the question remains how US below the HT could have caused these changes in the first place. While little is known about the processing of subliminal auditory stimuli, it is well documented that changes in neuronal activity can be detected in response to visual stimuli presented below the perceptual threshold [82, 83]. In addition, the fact that IFG has also been implicated in contrast-enhancement for cases of low auditory discriminability [77, 84, 85] and that auditory change detection can be recorded for consciously imperceptible stimulus differences [86, 87] suggests that auditory stimulation does not necessarily have to produce a clear and distinct hearing impression to affect the CNS. Importantly, a similar conclusion can be drawn from Ascone et al.'s study [16], who showed that prolonged exposure to inaudible US at SPLs well below the HT was associated with significant grey matter volume reductions in large frontal clusters (including bilateral IFG), even though participants were asleep during most of the exposure. Apart from that, it remains unclear why 9 out of 10 participants who reported hearing the stimuli during ATC also had a hearing impression during BTC, while reporting correctly that no stimulation had occurred when the sound source was switched off (NTC). In general, the fact that the intra-individual spread of HTs in the five participants who completed the HT assessment twice on two non-consecutive days only ranged between -1 and +5 dB SPL, indicates that the experimental setup operated with enough precision to reliably present sound above the HT and that stimulation during (n-)BTC (presented 10 dB below the HT) was low enough to account for day-to-day fluctuations in hearing acuity. In addition, it needs to be emphasized that the presence of scanner noise increases HTs rather than decreases them [88] and that changes in head- or earplug positioning would likely reduce the SPL in the ear canal,

since the ear plug position was optimized via a 8 kHz probe tone prior to the scan. We can therefore only speculate that participants may have found it implausible that we did not expose them to US when asking for a perceptual report and therefore guessed randomly (with a rate of 66% false estimates, which is close to a 50% probability of guessing).

Furthermore, even though several neuroimaging studies suggest that the hemodynamic response due to acoustic stimulation can be recorded at SPLs roughly corresponding to the participants' HTs [88, 89], we again found no evidence for auditory processing. As can be derived from the verbal reports, US during ATC with an average SPL of 111.3 dB SPL (average HT was at 106.3 dB SPL) was perceived as 'slightly unpleasant' (one participant perceived it as 'very unpleasant'), which not only indicates that stimulation was above the HT, but that there may be little leeway for further increases in SPL without putting participants at risk. In view of these findings, it seems rather unlikely that the lack of evidence for auditory processing can be attributed to weak stimulation or low sensitivity of MRI. Instead, even though there is little spectral overlap between the stimulus frequency used in this study (21.5 kHz) and scanner noise (highest spectral density of an EPI sequence is at around 1.4 kHz [90]), it is conceivable that ambient noise may have exerted a masking effect on the hemodynamic response, as has been described in previous studies using auditory stimuli in the typical hearing range [91–93]. Apart from that, it is known that mapping the tonotopic organization of PAC by means of fMRI is complicated by several additional factors [89]. For example, the functional architecture of PAC exhibits considerable inter-individual variability [94] and several studies have shown that neuronal response properties can be modulated via top-down attentional mechanisms [95–97]. It thus appears that future studies aimed at identifying the neural correlates of US perception would benefit from carefully considering those aspects, i.e. whether stimuli are actively attended or listened to passively, while also addressing inter-individual variability, for example by screening for US sensitivity as part of the recruitment process.

## Conclusion

The present study indicates that US can influence activity in the cognitive control network only when administered below the HT and that this effect is more pronounced the more unpleasant US during BTC compared to ATC is perceived. Previous studies have reported various subjective symptoms in response to US exposure, such as 'annoyance', 'inability to concentrate' [8, 15], 'vertigo' or 'tingling in the limbs' [1, 98]. The findings of this study add to the existing body of research, by suggesting that in addition to brain structure [16], US can also affect brain activity in areas involved in executive functions such attentional control and inhibition. Referencing evidence from various lines of research, the authors argued that bilateral IFG activation could reflect an increase in cognitive demand, mediated by attentional mechanisms in the presence of unpleasant and/or distractive stimuli. In addition, this study is the first to report brain-behavior effects associated with US exposure. The fact that higher bilateral IFG activation was associated with a decrease in RTs during n-BTC and that error-rates during n-BTC and n-ATC were lower the more unpleasant sound during BTC was perceived, suggests that bilateral IFG activation may have helped those participants who were affected more negatively by US to maintain their level of cognitive performance. However, the validity of this study is limited by the fact that verbal reports were not entirely consistent and that changes in the cognitive control network could not be related to auditory stimulation directly, since no evidence for sensory processing was obtained. We therefore propose that future studies revisit these issues, using different neuroscientific measurement tools in combination with a powerful in-ear sound source to further investigate the neural correlates of US under carefully controlled experimental conditions. Moreover, additional studies using inhibition (i.e., stop-

signal) as well as sustained and selective attention tasks are required to better understand the effects of US on cognitive processing and thus help updating existing guidelines and regulations for public US exposure.

## Author Contributions

**Conceptualization:** Christian Koch, Simone Kühn.

**Data curation:** Markus Weichenberger, Marion U. Bug.

**Formal analysis:** Markus Weichenberger, Marion U. Bug, Rüdiger Brühl, Simone Kühn.

**Funding acquisition:** Bernd Ittermann, Christian Koch.

**Investigation:** Markus Weichenberger, Marion U. Bug, Rüdiger Brühl.

**Methodology:** Markus Weichenberger, Marion U. Bug, Rüdiger Brühl, Christian Koch, Simone Kühn.

**Project administration:** Bernd Ittermann, Christian Koch, Simone Kühn.

**Resources:** Bernd Ittermann, Christian Koch, Simone Kühn.

**Software:** Markus Weichenberger, Rüdiger Brühl, Simone Kühn.

**Supervision:** Bernd Ittermann, Christian Koch, Simone Kühn.

**Validation:** Rüdiger Brühl, Christian Koch, Simone Kühn.

**Visualization:** Markus Weichenberger, Marion U. Bug.

**Writing – original draft:** Markus Weichenberger, Marion U. Bug, Simone Kühn.

**Writing – review & editing:** Markus Weichenberger, Marion U. Bug, Rüdiger Brühl, Bernd Ittermann, Christian Koch, Simone Kühn.

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
