## [Decision Letter · Decision Letter 0]

13 Apr 2022

PONE-D-21-19549Air-conducted ultrasound below the hearing threshold produces functional changes in the cognitive control network.PLOS ONE

Dear Dr. Weichenberger,

Thank you for submitting your manuscript to PLOS ONE. After careful consideration, we feel that your research has merit, but there are serious issues with the framing and interpretation of the results that have been raised by both reviewers of your manuscript.  Therefore, we conclude that this research does not yet meet PLOS ONE’s publication criteria. However, both reviewers were optimistic that a substantive revision might make your work publishable, therefore, we invite you to submit a revised version of the manuscript that addresses the points raised during the review process.

Reviewer 2 also asked that you address these points in your revision:

**SPECIFIC COMMENTS from Reviewer 2:**

Please amend the paper to address these, rather than keep the paper barely changed and explain back to the reviewer, because if the reviewer was unclear what was meant, the average reader might well be, and they will not see any of the clarifying material in the response to reviewer.

I would have thought the Editor might comment on whether it is appropriate to have so many acronyms in the Abstract. (Note from me regarding this comment:  The number of acronyms/initialisms used in the abstract may be addressed in the final revision stage of the manuscript, if the paper is deemed suitable for publication.  It would be wise to consider this issue in this revision to reduce your workload later)

p9. “HTs for pure tones with frequencies ranging from 8 to 26.5 kHz were measured monaurally (right ear) while participants lay in the scanner, in order to ensure that US audibility was preserved during the imaging process (background noise level was above the limit according to ISO 8253-1).” Were the HTs measured in an identical environment to the resting-state scanning condition, in terms of scanner noise etc? If so, why was the stimulus duration 1 sec during HT assessment and 2.75 sec during resting-state fMRI?

p.10 “Participants were not informed about the order in which the runs were conducted.” Were the testers aware of the order – i.e. double blind?

p. 10 “whereas the ATC and BTC runs were counterbalanced across participants.”

It could not be fully counterbalanced as there were an odd number of subjects. But admittedly any order effect is likely to be minor with at 8/7 imbalance. However the exact wording here cannot be correct.

p.11 “In each run, the 12 blocks were aligned with the stimulus protocol in the following order: n-NTC, n-BTC, n-ATC, n-NTC, n-BTC, n-ATC, n-ATC, n-BTC, n-NTC, n-ATC, n-BTC, n-NTC”.

What was the rationale for choosing this sequence?

p.11 “After each of the six runs, participants were asked ’Did you hear sounds during the last run?’”. Given that the scanner noise is loud, presumably they were also instructed to only answer “yes” for sounds other than the scanner noise.

p.14. “we contrasted whole brain data gathered during n-NTC vs. fixation”.

I do not know what “fixation” means here. I am guessing it is results of a pre-task scan? Please clarify in the paper.

p. 15 “To do so, we calculated pairwise contrasts between activations during the two stimulus conditions (n-ATC + n-BTC) and n-NTC, however no significant differences were found (p < 0.001, uncorrected)”. Does “n-ATC + n-BTC”

Does this mean that the two activations were first summed, and then contrasted with n-NTC, or that two contrasts were conducted: n-ATC vs n-NTC and n-BTC vs n-NTC? Further down, it seems that excitation of IFG for n-BTC> n-ATC (statistically significant). What about n-BTC vs n-NTC? This seems like a sensible planned comparison, given the hypothesis that “BTC cannot be detected and so would not be expected to lead to excitation? If the BTC excitation is causing some effect in the brain, does this show up in the BTC vs. NTC contrast?

p. 15 “To do so, we calculated pairwise contrasts between activations during the two stimulus conditions (n-ATC + n-BTC) and n-NTC, however no significant differences were found (p < 0.001, uncorrected)”.

Presumably this should be p>0.001 if it was not significant?

p.15 “However, when contrasting the two stimulation runs directly (n-BTC vs. n-ATC) we found a strong activation in bilateral inferior frontal gyrus (IFG, triangular part) only when US was presented below the HT (p < 0.001, cluster > 30)”

I don’t know what “cluster >30” means here, but it probably makes sense to an fMRI expert. Please clarify for the acousticians and audiologists who will be reading this.

p. 15. “after NTC, 11 out of 15 accurately stated that no stimulation had taken place, two were unsure and two reported that sounds had been presented.”

Not consistent with table 2 which states 3 were unsure.

p. 15. “Apart from that, paired t-test (two-tailed) revealed that stimulation during ATC (mean = -1.00, SD = 1.13) was rated significantly more unpleasant compared to BTC (mean = 0.07, SD = 0.70) (t(15) = 3.23, p = 0.006).”

How are there get 15 degrees of freedom in the t-statistic? If all 15 were included then you would have 14 dfs. But presumably only subjects who reported sounds in both ATC and BTC conditions who could be included – so at most 10, and possibly fewer, because it is stated that only subjects who reported sounds present then rated the pleasantness of the sound. And it could be <10, since the paired test can only be performed on subjects who were correct in both ATC and BTC conditions. It isn’t stated how many of the 10 who reported sounds present in ATC also reported them present in BTC. And also how many of these wrongly reported sounds present in NTC?

p. 15. “When participants were asked ‘Did you hear sounds during the last run?’ after the first n-back run, 7 out of 15 (one ‘unsure’)”

Since the run contained at four blocks ATC, why do you think the sounds were not detected?

p. 16. “While no correlations between resting-state brain data and verbal reports reached significance, we found that bilateral IFG activation during n-BTC vs. n-ATC was associated with verbal reports after resting-state in two ways: First, we found that the more pleasant a tone was perceived during ATC, the higher activation in bilateral IFG during ATC was (r = 0.602, p = 0.023).”

Should that be “higher activation in bilateral IFG during n-ATC vs n-BTC”?

p. 16. “While no correlations between resting-state brain data and verbal reports reached significance, we found that bilateral IFG activation during n-BTC vs. n-ATC was associated with verbal reports after resting-state in two ways: First, we found that the more pleasant a tone was perceived during ATC, the higher activation in bilateral IFG during ATC was (r = 0.602, p = 0.023).”

What are the degrees of freedom for the estimate of r? For verbal reports in the resting state, there were only 10 subjects who were asked about pleasantness, so presumably there were only 9 dfs for this test. R=0.602 is not significant with dfs=9 on a two-tailed test.

p.16. A number of different correlations are presented. Were these all a priori planned correlations? ATC pleasantness vs. n-ATC minus n-BTC IFG activation is presented, and also ATC pleasantness minus BTC pleasantness vs. n-ATC minus n-BTC IFG activation. Why these combinations? E.g. why not also BTC pleasantness vs. n-ATC minus n-BTC IFG activation? Other correlation coefficients are calculated involving for IFG activation, RTs, pleasantness rating, error rates. This leads to a large number of potential correlations and hence an inflated type 1 error rate unless there were selected planned comparisons.

p. 17. “First, we showed that the more unpleasant sound was perceived during BTC compared to ATC the higher signal strength in bilateral IFG was (although ATC was generally experienced as more uncomfortable).”

In the discussion, the authors don’t mention that they report a correlations between pleasantness in ATC and IFG activation in ATC (though I think the latter might be n-ATC vs. n-BTC). Is this also consistent with your hypothesis?

P 17. “WM” not defined. Presumably “working memory”.

Ethics statement and page 7. “The study was conducted according to the Declaration of Helsinki with approval of the ethics committee of the German Psychological Association (DGPs).” The authors should state the year of the Declaration of Helsinki (because it changes significantly with different versions).

REFERENCES

Cieslak M, Kling C and Wolff A (2020) Ultrasound exposure in a workplace and a potential way to improve its measurement methodology. 2020 IEEE International Workshop on Metrology for Industry 4.0 & IoT, 2020, pp. 172-176, doi: 10.1109/MetroInd4.0IoT48571.2020.9138223.Cieslak M, Kling C, Wolff A. (2021) Development of a Personal Ultrasound Exposimeter for Occupational Health Monitoring. International Journal of Environmental Research and Public Health. 18(24):13289. https://doi.org/10.3390/ijerph182413289Duck, F. and Leighton, T.G. (2018) Frequency bands for ultrasound, suitable for the consideration of its health effects. J. Acoust. Soc. Am. 144(4) 2490-2500 (doi: 10.1121/1.5063578)Dolder, C.N., Fletcher, M.D., Lloyd Jones, S., Lineton, B., Dennison, S.R., Symmonds, M., White, P.R. and Leighton, T.G. (2018) Measurements of ultrasonic deterrents and an acoustically branded hairdryer: Ambiguities in guideline compliance. J. Acoust. Soc. Am. 144(4), 2565-2574 (doi: 10.1121/1.5064279)Fletcher, M.D., Lloyd Jones, S., White, P.R., Dolder, C.N., Leighton, T.G. and Lineton, B. (2018a) Effects of very high-frequency sound and ultrasound on humans. Part I: Adverse symptoms after exposure to audible very-high frequency sound. J. Acoust. Soc. Am. 144(4), 2511-2520 (doi: 10.1121/1.5063819)Fletcher, M.D., Lloyd Jones, S., White, P.R., Dolder, C.N., Leighton, T.G. and Lineton, B. (2018b) Effects of very high-frequency sound and ultrasound on humans. Part II: A double-blind randomized provocation study of inaudible 20-kHz ultrasound. J. Acoust. Soc. Am. 144(4), 2521-2531 (doi: 10.1121/1.5063818)Fletcher, M.D., Lloyd Jones, S., White, P.R., Dolder, C.N., Lineton, B. and Leighton, T.G. (2018c) Public exposure to ultrasound and very high-frequency sound in air. J. Acoust. Soc. Am. 144(4), 2554-2564 (doi: 10.1121/1.5063817)Leighton, T.G., (2017). Comment on ‘Are some people suffering as a result of increasing mass exposure of the public to ultrasound in air? Proc. Math. Phys. Eng. Sci. 473, 20160828Leighton, T.G. (2018) Ultrasound in air - Guidelines, applications, public exposures, and claims of attacks in Cuba and China. J. Acoust. Soc. Am. 144(4) 2473-2489 (doi: 10.1121/1.5063351).Leighton, T. G. (2020) Ultrasound in air - Experimental studies of the underlying physics are difficult when the only sensors reporting contemporaneous data are human beings. Physics Today, 73(12), 39-43 (doi: 10.1063/PT.3.4634).Leighton, T. G., Currie, H. A. L., Holgate, A., Dolder, C. N., Lloyd Jones, S., White, P. R. and Kemp, P. S. (2020a). Analogies in contextualizing human response to airborne ultrasound and fish response to acoustic noise and deterrents, Proceedings of Meetings on Acoustics (POMA) 37, 010014 (doi: 10.1121/2.0001260).Leighton, T. G., Lineton, B., Dolder, C. N. and Fletcher, M. D. (2020b) Public Exposure to airborne ultrasound and Very High Frequency sound. Acoustics Today, 16(3), 17-26 (doi: 10.1121/AT.2020.16.3.17).Lubner R. J., Kondamuri N. S., Knoll R. M., Ward B. K., Littlefield P. D., Rodgers D., Abdullah K. G., Remenschneider A. K. and Kozin E. D. (2020) Review of Audiovestibular Symptoms Following Exposure to Acoustic and Electromagnetic Energy Outside Conventional Human Hearing. Frontiers in Neurology, 11, Article 234, DOI=10.3389/fneur.2020.00234Maccà, I., Scapellato, M. L., Carrieri, M., Maso, S., Trevisan, A. and Bartolucci, G. B. (2015) High-frequency hearing thresholds: effects of age, occupational ultrasound and noise exposure. Int. Arch. Occup. Environ. Health 88, 197–211. (doi:10.1007/s00420-014-0951-8)Mapp, P. A. (2018) Potential audibility of ultrasonic signal monitoring of Public Address and Life Safety Sound Systems, J. Acoust. Soc. Am. 144(4), 2539–2547Scholkmann, F. (2019) Exposure to High-Frequency Sound and Ultrasound in Public Places: Examples from Zurich, Switzerland. Acoustics 1, 816 (2019).Ueda, M., Ota, A. and Takahashi, H. (2014) Investigation on high-frequency noise in public space. Internoise 2014, Melbourne Australia, 16–19 November 2014, 7 p.Ullisch-Nelken C., Wolff, A., Schöneweiß, R. and Kling, C. (2017), A measurement procedure for the assessment of industrial ultrasonic noise, Proceedings of the 25th International Congress on Sound and Vibration, Vol. 4, pp. 2433–2438, Curran Associates, Red Hook, NY, USAvan Wieringen, A. and Glorieux, C. (2018) Assessment of short-term exposure to an ultrasonic rodent repellent device,” J. Acoust. Soc. Am. 144(4), 2501–2510.

We look forward to receiving your revised manuscript.

Kind regards,

Brenton G. Cooper, Ph.D.

Academic Editor

PLOS ONE

Journal Requirements:

5. Please include your tables as part of your main manuscript and remove the individual files. Please note that supplementary tables (should remain/ be uploaded) as separate "supporting information" files"

Reviewers' comments:

Reviewer's Responses to Questions

**Comments to the Author**

1. Is the manuscript technically sound, and do the data support the conclusions?

Reviewer #1: No

Reviewer #2: Partly

2. Has the statistical analysis been performed appropriately and rigorously? 

Reviewer #1: Yes

Reviewer #2: I Don't Know

3. Have the authors made all data underlying the findings in their manuscript fully available?

Reviewer #1: Yes

Reviewer #2: No

4. Is the manuscript presented in an intelligible fashion and written in standard English?

Reviewer #1: Yes

Reviewer #2: Yes

5. Review Comments to the Author

Reviewer #1: The authors investigated the effect of high-frequency sounds on a visuo-spatial working memory task in human subjects. Although the study fails to demonstrate acoustic evoked activity, the authors describe an effect of sub-threshold high-frequency stimulation on brain network activity.

As there are no line numbers included in the manuscript, which is unfortunate, I will first only provide general comments here:

General comments:

My major issue with the current manuscript is the use of the word ultrasound, which is confusing in the context of this article. Technically, yes it is ultrasound as defined as frequencies above 20kHz. However, the definition of ultrasound is in that sense arbitrary, because ultrasound has also the connotation of being outside the human hearing range. The frequencies studied in this manuscript can in principle be heard by human subjects, even though they lie at the extrem higher end of the human hearing range (which is also demonstrated by the authors, by determining hearing thresholds). The use of ultrasound now leads to confusion, when ultrasound is used in the sense, that it refers to sounds outside the human hearing and audible range. This creates a grey zone, and it is unnecessarily misleading. The clarity of the manuscript would profit a lot, if this would be made rephrased, better defined and better distinguished, in the title, abstract, and introduction.

My second issue, closely related to the first, is that the use of US this article suggests that US (> 20 kHZ) is something fundamentally different as sound in the hearing range. Again this is misleading and not helpful. It is indeed interesting to study the effects of extremely high-frequencies on subjective experience, but I suggest to talk about high-frequencies in the hearing spectrum and not ultrasound. At least this needs to be pointed out much clearer. The study simply refers to high-frequency sounds, which lie at the border of the possible audible range (as demonstrated in the study). This is not new, and it is known that the human hearing audible range can reach 20kHz and above, especially in younger age groups. (e.g Rodriguez et al. 2014). Nobody debates this. As this study however is concerned with simply high-frequency sounds in the upper human audible range, I advise the authors to use a description, that better fits these facts. Especially, at least is important to point out that the study concerns US just at the lowest lower border, in contrast for example to ultrasound way above (e.g. >40 kHz). Further it needs to be made clear that the US distinction is somewhat arbitrary, with relation to possible human hearing thresholds. The manuscript should be modified accordingly.

The third issue, as devices operating with US are taken as a motivation of the study, it would be helpful to provide specific examples at which specific US frequency ranges the devices work (because US can mean many things, from only a little above 20000 Hz or 100kHz). Then it becomes clearer how this relates to the results in the manuscript. Ultrasound has a large range, and the wording in the title implies that the results would hold true for many US frequencies.

Further, it is not clearly enough described, how US can be perceived other that the cochlea, and especially how sub-threshold sounds! can have an effect on brain network activity. This needs to be motivated and explained more clearly, and the expected sensory path should be better described, as in the distinction of the normal auditory pathway via the hair cells, in contrast to an alternative path for US (via bone conduction, etc.) that is hypothesised in some studies. The title implies that the authors show mechanisms of US that can work via non-classical sensory pathways that can be detrimental to cognition. But it is not sufficiently explained how that actually should work, and whether this is not simply an artefact.

Reviewer #2: Air-conducted ultrasound below the hearing threshold produces functional changes in the cognitive control network. 2022

Markus Weichenberger, Marion Bug, Rüdiger Brühl, Bernd Ittermann, Christian Koch, Simone Kühn.

OVERALL

This paper reports a difficult experiment in an important field, with a number of interesting findings (such as the low scatter between individuals in hearing threshold, the fact that fMRI failed to detect a response when individuals heard a sound etc.).

However, it is let down by its placement in the wider context, both in terms of why it is important society looks at this now, and what previous studies have shown: the few statistically significant data (in this field dominated by subjective accounts) are not discussed.

An important finding appeared to be downplayed and would be missed by readers. As with the MEG data of Fujioka et al. (2002) here no response could be detected despite the fact that some of the subjects reporting hearing the stimulus. If true, that suggests these techniques are not sufficiently sensitive to warrant the assumption that they make infallible detectors for threshold studies into the response of humans to US. Would you agree? That would seem to be key given that the tendency of those outside of science is to see fMRI images and believe they represent the gold standard for detection of brain activity.

CONTEXT

The introduction opens with “Technological progress and urbanization significantly contributed to the fact that nowadays air-conducted sound in the ultrasonic frequency range (> 20,000 Hz, US) represents an integral part of our daily stimulus environment”. However, no evidence whatsoever is presented to back up this claim. This is particularly surprising as surely the impetus for study in this area would be the discovery of ultrasound exposing the public without their knowledge in public places (Leighton, 2016), which has been detected in EU and UK (Fletcher et al., 2018c), Japan (Leighton et al. 2020a), in US schoolrooms (Leighton, 2020; Leighton et al, 2020b) and in domestic products (Dolder et al., 2018). Of course ultrasonic deterrents have been emitting into air for decades, but their contributions to ultrasonic exposures in public spaces was unappreciated (Leighton, 2016; van Wieringen and Glorieux, 2018). Similarly, industrial exposures had been appreciated for decades, and drove the institution of guidelines for exposure (Ullisch-Nelken et al., 2017).

The paper does not give this context in the Introduction. Without it, the reader will struggle to appreciate why it is important that works on this topic be published. It is in the authors’ interests to explain FROM THE START the newly-emerging field of public exposures, and so explaining the situation that , in the eyes of their readers, will make the more compelling case for their research.

By only mentioning public exposures as an addendum, postscript to their long list of references on industrial research, the authors lead their case for publication with the topic of industrial exposures, for which there is a long history of research, and in which the sources of ultrasound are well known. This is clear when they say ‘less attention is paid to the fact that US is also emitted by an increasing number of technical devices, such as motion detectors, loudspeakers, or cleaning tools, which often results in people being exposed to such frequencies on a daily basis without noticing. A number of studies on US in occupational settings suggests that there may be a relationship between sound exposure and the occurrence of subjective symptoms, such as hearing threshold shifts, nausea, dizziness, migraine, fatigue, and tinnitus (Skillern, 1965; Acton & Carson, 1967; Acton, 1974, 1983; Von Gierke & Nixon, 1976; Crabtree & Forshaw, 1977; Herman & Powell, 1981; Damongoet & André, 1988).’ By LEADING with public exposures, they will make a much stronger case for why It is important for their work to be published. By explaining to the reader that this includes schools, for example, the reader will understand the imperative for such studies.

COMPARISON WITH OTHER STUDIES

The papers fails to put its findings in the context of other studies. The paper states ‘Several studies have demonstrated that humans are capable of perceiving US, yet little is known as to how such sounds are processed and whether adverse health effects might be associated with US exposure’ (Abstract, line 2). Furthermore, on page 3 the authors say:

“Meanwhile, initial studies on the effects of US emitted in public spaces also suggest that such sounds can be perceived as ’noisy’, ’uncomfortable’ and even as causing ’a headache or an earache’ (Ueda et al., 2014, Leighton, 2016, 2017) and devices with sound pressure levels (SPLs) of up to 147 dB, which apparently match, and in some cases exceed those measured in the manufacturing industries are now commercially available (Grigor’eva, 1966; Acton, 1968; Knight, 1968; EARS Project, 2015). Apart from the fact that the subjective symptoms attributed to US are very diverse and unspecific, there are a number of other issues that make it difficult to gain a better understanding of such associations.“

This gives the unfortunate impression to the reader that there are only vague subjective descriptions of effects. It ignores the statistically significant effects proven in some studies.

It is important to put this study in the context of the statistically significant conclusions on this that can be found in the literature. Given there are barely any statistically significant studies, it is important to include them. This is particularly so, because so many studies were contaminated by high levels of sound at lower frequencies. Macca et al compared a cohort of industrial workers who had been exposed to high levels of both ultrasound and sound in an industrial setting, with workers who had only been exposed to ultrasound. Macca et al reported statistically significant effects (although their statistical calculations were erroneous, and needed reworking by Leighton (2016) to determine which effects were statistically significant).

In contrast, Fletcher et al (2018a and b) got around the problem of contamination of the sound field by lower frequency noise, by excluding it. Fletcher et al. (2018a) showed statistically significant effects for ultrasound that could be perceived. Although they did not show it for ultrasound that could not be perceived (2018b) they warn that ethical guidelines meant that the durations and SPLs to which they could expose their subjects, were less than those they might encounter in a public place, and so the extrapolation of their results to infer ‘ultrasound at levels below the hearing threshold cannot harm you’ would be an erroneous inference. Of particular note, both students (Fletcher et al., 2018a and b) separated out subjects who had previously complained of adverse effects from what they believed to be ultrasound in public places (the ‘‘symptomatic’’ group), from subjects who had not (the ‘‘asymptomatic’’ group).

I feel it is important to explain how Fletcher et al introduced the innovation of separating (the ‘‘symptomatic’’ group from the ‘‘asymptomatic’’ group: this is a good experimental way of overcoming the wide range in susceptibilities seen in the population, especially when the number of subject tested (as in the submitted manuscript) is small.

Fletcher et al. (2018a) showed statistically significance for 'annoyance' and 'inability to concentrate', but was statistically proven by Fletcher et al. (2018a) and illustrated in a real-world case studies by Leighton (2020) and Leighton et al., (2020c) and Ueda (2014). Fletcher et al. (2018a) also found statistically significant galvanic skin responses (GSRs) in the ‘symptomatic’ group.

Maccà et al. (2015) showed a statistically significant increases in asthenia (loss or lack of bodily strength) and vertigo, and in recalculating their data, Leighton (2016) confirmed these findings but also showed a statistically significant increase in ‘tingling in the limbs’.

It is vital the authors properly review these studies or the reader will miss the imperative nature of the need to research and publish in this area.

The first line of the CONCLUSIONS needs to reflect this small but important body of statistically significant findings.

Furthermore the phrase “To the authors’ knowledge, this study is the first to demonstrate that air-conducted US is capable of producing changes in functional brain activity” is inappropriate. I think, at most, the authors showed evidence of this, but what they present falls short of a clear “demonstration”.

SENSITIVITY and CONTROLS

Surely it is very important that the techniques used in this paper fail to register an effect, when patients report hearing the sound. This suggests that both MEG (Fujioka et al. 2002) and fMRI (this paper) lack the sensitivity to be used as sensory response threshold detectors for high frequency acoustics and ultrasonic exposure. This point should be made here. And in the first line of the Discussion.

The paper leaves open the questions over whether the same can be said of the EEG data of Nittono (2020) et al. it is reported that no differences were detected – was it statistically tested? Could they hear the signals? In all these studies, did these past subjects actually hear the signals, or could they have been responding to something else? Please clarify.

In the same paragraph, the authors are coy as to why they are repeating the work of Kühler et al. (2019). It is vital that they are honest here. Kühler et al. (2019) used an inappropriate control. In the light of this, the current paper should not repeat the conclusions of that paper (‘the authors also found no evidence for auditory processing at frequencies above 14 kHz’) as this paper has already been cited by those who read it to mean human could not hear high frequencies. The fact that the control was inappropriate, and the point discussed here that MEG and fMRI are insufficiently sensitive for threshold detection, make it misleading to include text that would lead readers to believe that use of MEG and fMRI prove humans do not detect these frequencies.

The authors in the conclusion, “Whereas previous studies established rather unspecific associations between US exposure and a number of subjective symptoms (such as ‘annoyance’ or ‘discomfort’)”. This is not true given the work of Fletcher et al (2018a and b) and Macca et al (with stats re-run by Leighton 2016).

DETAILED COMMENTS.

Abstract line 1. The point has been established in the literature (Leighton, 2017, 2018; Scholkmann, 2019; van Wieringen and Glorieux, 2018; Lubner et al., 2020; Paxton et al., 2018; Fletcher et al., 2018a,b,c; Dolder et al, 2018; Duck and Leighton, 2018) that, because the third octave bands that is centered on 20 kHz, extends down to 17.8 kHz, all the guidelines for Maximum Permissible Levels (MPLs) set for 20 kHz, extend down to 17.8 kHz, making (by default) the lower limit of the ultrasonic band 17.8 kHz, not 20 kHz. Cieslak et al (2020, 2021) acknowledge the argument to set the lower limit of ultrasound at 17.8 kHz, but comply with German tradition of setting ti at 16 kHz, Given the 17.8 kHz has logical reasoning behind it, and beings in line most of the previous guidelines (which could not reasonable state 20 kHz as the limit whilst regulating down to 17.8 kHz when setting levels for ultrasonics exposure) I recommend you use that in the definition.

Page 16. “Participants were neutral regarding the effects of US during the first run, although

verbal reports varied considerably (mean = 0.00, SD = 1.51, min = -4, max = +3).” Is this strictly accurate? It would imply all subjects were neutral. So you mean the average response came out as neutral but that only a minority of subjects actually had a neutral response? This point was not clear and greater clarity would be appreciated.

EXPERIMENTAL

This is a difficult experiment, the need to reduce metal clearly making the experiment challenging. Nevertheless, the experiment was well-conducted, with good protocols, equipment and measurement.

NEED FOR FURTHER DISCUSSION on ‘BELOW THRESHOLD’

Some of the other results are surprising. The main one is that in the BTC condition, where the US-tone is presented at 10 dB below the previously measured threshold, most subjects said (correctly) they could detect the sound, while in the NTC condition where no sound was presented most said (again correctly) that they could not hear any sound. So there is an explained discrepancy as to why they could (seemingly) hear the sound during the “resting state scanning” but not when their threshold was assessed. The authors do not really discuss why this might be, though they rule out test-retest reliability of the hearing threshold which was better than +/- 5 dB. Also, in the first of the three n-back scan runs, half the subjects did not say they could hear any tones, even when there were quite a few presented at 5 dB above threshold. But then in the next two cases they (mostly) could hear the sounds (as expected).

It is not clear whether this anomaly can be explained due by differences in the test set up between threshold measurements and scanning, or some odd effect of the order of testing (I don't know what that might be). There needs to be clarity on this.

The authors appear to see differences in brain activity (bilateral IFG) between ATC and BTC stimulation, but it is not possible to tell from the paper whether they also see a difference between BTC and NTC (i.e. below threshold and no stimulus conditions) or ATC vs. NTC because they are unclear on this. The authors appear to claim this shows that the BTC condition is causing some effect in the brain, but it is currently impossible for the reader to tell whether this is valid. The authors found a difference in IFG between ATC vs BTC, but could it be that this was an inverse effect that was caused by the ATC case (relative to NTC)? I would have thought that the obvious hypothesis to test would be that the BTC condition gives the same response as the NTC condition (since in both cases there is no audible US).

So, when talking about the “below threshold” condition, the authors need to be careful. Indeed, I think it is not correct to call this ‘below threshold’ without qualification – because the reader will read the paper thinking this refers to levels below the threshold for inciting a response, but these ‘below threshold’ levels clearly are not always below threshold!

Options might be that the detection sensitivity of the experiment is not sufficient.

Or there might be large variations in the level reaching the ear canal based on fittings or movement of the fittings within in the ear canal or something like that because the sound is at such a high frequency.

Given how much text the authors devote to the DISCUSSSION, they really don’t discuss this major issue much.

There may also be a problem with the correlations because as I understand it the authors only take a pleasantness rating when the person reports being able to perceive the sounds. It’s odd to read because the sound is “below threshold” yet the subjects only rate it’s pleasantness when they perceive it… if they can perceive it, then it’s obviously not below threshold…

This identifies one limitation of the experiment vs the real world, which is that because of their rating scales (getting subjects to rate pleasantness and influence of the sounds), the authors are really getting the participants to focus in on the sounds each time, which they likely wouldn’t necessarily be doing with real-world exposure. Would it not be a valid criticism that these neural responses are quite sensitive to whether the subject is paying attention? If so, this needs discussion.

STATISTICS

Also, some of the statistical analysis is odd, because in some places they appear to use all 15 subjects, when there were only at most 10 who should have had valid results (and possibly fewer as they could only include people who “heard” the tone in both the ATC and BTC conditions). The authors should revisit this point and explain themselves, or correct the analysis if necessary.

The authors conduct quite a few statistical tests, where it is not clear whether these were planned before hand or not; this could inflate their false-positive rate if they were unplanned.

These are definitely areas where the authors need to provide clarification.

The character count would not let me include the remainder of my review and so I have sent that separately to the editor, so please contact the editor if you do not receive the rest.

6. PLOS authors have the option to publish the peer review history of their article (what does this mean?). If published, this will include your full peer review and any attached files.

Reviewer #1: No

Reviewer #2: No

---

## [Author Response · Author response to Decision Letter 0]

5 Oct 2022

The document "Response to reviewers" is now part of the attached files.

Reviewer #1:

The authors investigated the effect of high-frequency sounds on a visuo-spatial working memory task in human subjects. Although the study fails to demonstrate acoustic evoked activity, the authors describe an effect of sub-threshold high-frequency stimulation on brain network activity. 

As there are no line numbers included in the manuscript, which is unfortunate, I will first only provide general comments here:

• (1) Author’s response: Thank you for bringing this to our attention. Line numbers have now been included in the revised version of the manuscript. However, due to the fact that Microsoft Word doesn’t accurately display continuous line numbers when reviewing a document, line numbers will restart each page.

General comments:

My major issue with the current manuscript is the use of the word ultrasound, which is confusing in the context of this article. Technically, yes it is ultrasound as defined as frequencies above 20kHz. However, the definition of ultrasound is in that sense arbitrary, because ultrasound has also the connotation of being outside the human hearing range. The frequencies studied in this manuscript can in principle be heard by human subjects, even though they lie at the extrem higher end of the human hearing range (which is also demonstrated by the authors, by determining hearing thresholds). The use of ultrasound now leads to confusion, when ultrasound is used in the sense, that it refers to sounds outside the human hearing and audible range. This creates a grey zone, and it is unnecessarily misleading. The clarity of the manuscript would profit a lot, if this would be made rephrased, better defined and better distinguished, in the title, abstract, and introduction.

My second issue, closely related to the first, is that the use of US this article suggests that US (> 20 kHZ) is something fundamentally different as sound in the hearing range. Again this is misleading and not helpful. It is indeed interesting to study the effects of extremely high-frequencies on subjective experience, but I suggest to talk about high-frequencies in the hearing spectrum and not ultrasound. At least this needs to be pointed out much clearer. The study simply refers to high-frequency sounds, which lie at the border of the possible audible range (as demonstrated in the study). This is not new, and it is known that the human hearing audible range can reach 20kHz and above, especially in younger age groups. (e.g Rodriguez et al. 2014). Nobody debates this. As this study however is concerned with simply high-frequency sounds in the upper human audible range, I advise the authors to use a description, that better fits these facts. Especially, at least is important to point out that the study concerns US just at the lowest lower border, in contrast for example to ultrasound way above (e.g. >40 kHz). Further it needs to be made clear that the US distinction is somewhat arbitrary, with relation to possible human hearing thresholds. The manuscript should be modified accordingly.

(2) Author’s response: Thank you very much for the feedback. We realized that the way, we used the term ultrasound and introduced the reader to the debate about the audibility of US has led to some confusion, which is why we decided to rewrite large parts of the introduction. Reviewer #2 also pointed out the arbitrary nature of defining US as sound frequencies > 20 kHz and referred us to an article by Leighton (2017), who showed that current ultrasonic regulation guidelines extend down to 17.8 kHz, which is the lower frequency limit of the third octave band centered in 20 kHz. In line with a number of recent studies (f.e. Wieringen and Glorieux, 2018; Fletcher et al., 2018a,b,c; Dolder et al, 2018; Paxton et al., 2018), we therefore propose to adopt this definition instead and use it in the Abstract (p1, line 3) and again in the introduction (p2, line 4) of the revised manuscript. To better inform the reader of which sound frequencies the study is concerned with, we also specified the range for which hearing thresholds in the US frequency spectrum have so far been determined (p2, lines 17 – 22). As you noted, the fact that human hearing extends into the US frequency spectrum is well known among readers interested in audiology, but we believe that a wider audience may benefit, if this was also pointed out in the article. We therefore challenge the idea held among large parts of the public that US is categorically outside of the human hearing range (p2, line 16 – 17), followed by direct reference to the existing literature on the matter. We hope this positively contributes to the clarity of the paper and helps reduce confusion due to unclear terminology.

The third issue, as devices operating with US are taken as a motivation of the study, it would be helpful to provide specific examples at which specific US frequency ranges the devices work (because US can mean many things, from only a little above 20000 Hz or 100kHz). Then it becomes clearer how this relates to the results in the manuscript. Ultrasound has a large range, and the wording in the title implies that the results would hold true for many US frequencies.

• (3) Author’s response: Thank you for the important remark. As part of the rewrite of the introduction, we have now referenced studies, in which acoustic field measurements were conducted in various public places to assess ultrasonic noise pollution and also included specific examples of technical devices (i.e. public address voice alarm (PAVA) systems and ultrasonic pest repellents) that are considered to be the biggest contributors of US in the public domain (p2, lines 9 – 16).

Further, it is not clearly enough described, how US can be perceived other that the cochlea, and especially how sub-threshold sounds! can have an effect on brain network activity. This needs to be motivated and explained more clearly, and the expected sensory path should be better described, as in the distinction of the normal auditory pathway via the hair cells, in contrast to an alternative path for US (via bone conduction, etc.) that is hypothesised in some studies. The title implies that the authors show mechanisms of US that can work via non-classical sensory pathways that can be detrimental to cognition. But it is not sufficiently explained how that actually should work, and whether this is not simply an artefact.

• (4) Author’s response: We thank the reviewer for bringing this to our attention. First of all, we realized that the repeated references to bone-conducted US perception and hypotheses regarding the signal transmission of bone-conducted US in the introduction did not provide critical information for understanding the article and probably lead to confusion, which is why we suggest to remove these references from the revised manuscript (with the exception of Hosoi et al.’s study (1998) who first demonstrated that bone-conducted US can lead to activation of the primary auditory cortex). Instead, we tried to emphasize more strongly that regardless of whether US is air- or bone-conducted, hearing in the US frequency spectrum most likely involves processing in the cochlea and should therefore be regarded as an auditory sensation (p3, lines 30 – 34). Second, apart from the hypothesis by Leighton (2016) and his reference to Job et al.’s work (2011), we are not aware of any other studies suggesting the existence of a different (in this case somatosensory) processing route for (inaudible) US and since we did not find activation of primary somatosensory areas in response to these stimuli, we also did not speculate about potential down-stream targets of an alternative processing route in the discussion. While we think it is appropriate to briefly touch on this hypothesis in the introduction, we tried to make it clear that the measured brain effect rather points towards the engagement of attentional mechanisms in response to auditory stimulation. Given the high statistical significance of our main effect (p < 0.001, cluster > 30) as well as the fact that IFG is critically important in attentional control and auditory change detection, we believe it is very unlikely that this effect represents an artifact. In the revised manuscript, we also referenced a recent longitudinal study by Ascone et al. (2021), who reported brain effects in response to inaudible US, thus providing additional evidence for the claim that US below the HT can exert an influence on the CNS (p3 lines 15 – 28). Moreover, while little is known about the processing of subliminal auditory stimuli, it is well documented that changes in neuronal activity can also be detected in response to visual stimuli presented below the perceptual threshold (e.g., see Brooks et al. 2012; Meneguzzo et al. 2014) (p20 lines 8 – 11).

Reviewer #2: 

1. OVERALL

This paper reports a difficult experiment in an important field, with a number of interesting findings (such as the low scatter between individuals in hearing threshold, the fact that fMRI failed to detect a response when individuals heard a sound etc.).

However, it is let down by its placement in the wider context, both in terms of why it is important society looks at this now, and what previous studies have shown: the few statistically significant data (in this field dominated by subjective accounts) are not discussed. An important finding appeared to be downplayed and would be missed by readers. As with the MEG data of Fujioka et al. (2002) here no response could be detected despite the fact that some of the subjects reporting hearing the stimulus. If true, that suggests these techniques are not sufficiently sensitive to warrant the assumption that they make infallible detectors for threshold studies into the response of humans to US. Would you agree? That would seem to be key given that the tendency of those outside of science is to see fMRI images and believe they represent the gold standard for detection of brain activity.

• (1) Author’s response: It is indeed surprising that neither fMRI nor MEG could detect brain activation in auditory brain regions, especially since these methods complement each other very well in terms of their strengths and weaknesses (i.e. high spatial and low temporal resolution and vice versa). However, it needs to be emphasized that the drastically different experimental conditions in the few available studies touched on in the manuscript (i.e. Fujioka et al. 2002; Kühler et al., 2019; Nittono, 2020) significantly impair their comparability and do not allow us to make general statements about the status of different measurement tools for detecting neural activity in response to airborne US (p4, lines 27 – 34; p5, lines 1 – 2). Within the scope of this article, we tried to focus primarily on the use of fMRI and suggest that the lack of activation in the primary auditory cortex during US above the HT can most likely be attributed to scanner noise contaminating our imaging data. In principle, we have no reason to assume that fMRI per se is not sensitive enough to detect neural activation in response to US, especially once it is perceived and experienced as unpleasant or even painful (p4, lines 30 – 33; p21 lines 1 – ).

• In the case of Fujioka et al.‘s MEG & fMRI study (2002), they showed that no significant changes in brain activity were observed when stimulation exceeded 14 kHz, even though some participants reported they still had a hearing impression when being exposed to sounds of up to 20 kHz, however sounds were applied via a loudspeaker over 1 meter away from the participants’ ears at a fixed SPL of only 60 dB. Based on their findings, the author suggested that such high frequencies are not represented in the tonotopy of the auditory cortex. We believe this conclusion was drawn somewhat prematurely, since it has been shown repeatedly that SPLs usually need to be much higher to reliably cause perception in the US frequency band. Moreover, Hosoi et al. (1998) already showed that by presenting US via bone-conduction, activity in the auditory cortex could be detected via MEG at frequencies of up to 40 kHz. 

• In the case of Nittono’s EEG study (2020), the author only stated that participant’s high-frequency auditory thresholds were between 14,000 and 19,000 Hz (M = 17,316 Hz) and it is not clear, how these thresholds were determined. Given the information provided in the article, we only know that the 2 test-stimuli (i.e. the 11- and the 22-kHz high-cut sound) were again delivered via 2 loudspeakers placed 1.2 m in front of the participants with a SPL of 62 dB. Therefore, it is highly unlikely that participants had a hearing impression of the 22 kHz tones, given the fact that SPLs of around 100 dB SPL are required to produce auditory perception in this frequency range, even when administering sound via an in-ear-device.

• Regarding the study of Kühler et al (2019), the authors are not quite sure what the reviewer means by “the control was inappropriate” as well as “this paper has already been cited by those who read it to mean human could not hear high frequencies” in comment 4. 

2. CONTEXT

The introduction opens with “Technological progress and urbanization significantly contributed to the fact that nowadays air-conducted sound in the ultrasonic frequency range (> 20,000 Hz, US) represents an integral part of our daily stimulus environment”. However, no evidence whatsoever is presented to back up this claim. This is particularly surprising as surely the impetus for study in this area would be the discovery of ultrasound exposing the public without their knowledge in public places (Leighton, 2016), which has been detected in EU and UK (Fletcher et al., 2018c), Japan (Leighton et al. 2020a), in US schoolrooms (Leighton, 2020; Leighton et al, 2020b) and in domestic products (Dolder et al., 2018). Of course ultrasonic deterrents have been emitting into air for decades, but their contributions to ultrasonic exposures in public spaces was unappreciated (Leighton, 2016; van Wieringen and Glorieux, 2018). Similarly, industrial exposures had been appreciated for decades, and drove the institution of guidelines for exposure (Ullisch-Nelken et al., 2017). The paper does not give this context in the Introduction. Without it, the reader will struggle to appreciate why it is important that works on this topic be published. It is in the authors’ interests to explain FROM THE START the newly-emerging field of public exposures, and so explaining the situation that , in the eyes of their readers, will make the more compelling case for their research.

By only mentioning public exposures as an addendum, postscript to their long list of references on industrial research, the authors lead their case for publication with the topic of industrial exposures, for which there is a long history of research, and in which the sources of ultrasound are well known. This is clear when they say ‘less attention is paid to the fact that US is also emitted by an increasing number of technical devices, such as motion detectors, loudspeakers, or cleaning tools, which often results in people being exposed to such frequencies on a daily basis without noticing. A number of studies on US in occupational settings suggests that there may be a relationship between sound exposure and the occurrence of subjective symptoms, such as hearing threshold shifts, nausea, dizziness, migraine, fatigue, and tinnitus (Skillern, 1965; Acton & Carson, 1967; Acton, 1974, 1983; Von Gierke & Nixon, 1976; Crabtree & Forshaw, 1977; Herman & Powell, 1981; Damongoet & André, 1988).’ By LEADING with public exposures, they will make a much stronger case for why It is important for their work to be published. By explaining to the reader that this includes schools, for example, the reader will understand the imperative for such studies.

• (1) Author’s response: Thank you for the feedback. Some of these concerns have also been raised by Reviewer #1 and we realized that in the previous draft of the manuscript, the reader was not introduced properly to the discourse about public US exposure, since many important studies were either not available at the time of writing the introduction or were not selected when reviewing the existing literature. We therefore decided to completely rewrite large parts of the introduction and put a much stronger emphasize on the available data regarding public US exposure. 

3. COMPARISON WITH OTHER STUDIES

The papers fails to put its findings in the context of other studies. The paper states ‘Several studies have demonstrated that humans are capable of perceiving US, yet little is known as to how such sounds are processed and whether adverse health effects might be associated with US exposure’ (Abstract, line 2). Furthermore, on page 3 the authors say: “Meanwhile, initial studies on the effects of US emitted in public spaces also suggest that such sounds can be perceived as ’noisy’, ’uncomfortable’ and even as causing ’a headache or an earache’ (Ueda et al., 2014, Leighton, 2016, 2017) and devices with sound pressure levels (SPLs) of up to 147 dB, which apparently match, and in some cases exceed those measured in the manufacturing industries are now commercially available (Grigor’eva, 1966; Acton, 1968; Knight, 1968; EARS Project, 2015). Apart from the fact that the subjective symptoms attributed to US are very diverse and unspecific, there are a number of other issues that make it difficult to gain a better understanding of such associations.“

This gives the unfortunate impression to the reader that there are only vague subjective descriptions of effects. It ignores the statistically significant effects proven in some studies. It is important to put this study in the context of the statistically significant conclusions on this that can be found in the literature. Given there are barely any statistically significant studies, it is important to include them. This is particularly so, because so many studies were contaminated by high levels of sound at lower frequencies. Macca et al compared a cohort of industrial workers who had been exposed to high levels of both ultrasound and sound in an industrial setting, with workers who had only been exposed to ultrasound. Macca et al reported statistically significant effects (although their statistical calculations were erroneous, and needed reworking by Leighton (2016) to determine which effects were statistically significant).

In contrast, Fletcher et al (2018a and b) got around the problem of contamination of the sound field by lower frequency noise, by excluding it. Fletcher et al. (2018a) showed statistically significant effects for ultrasound that could be perceived. Although they did not show it for ultrasound that could not be perceived (2018b) they warn that ethical guidelines meant that the durations and SPLs to which they could expose their subjects, were less than those they might encounter in a public place, and so the extrapolation of their results to infer ‘ultrasound at levels below the hearing threshold cannot harm you’ would be an erroneous inference. Of particular note, both students (Fletcher et al., 2018a and b) separated out subjects who had previously complained of adverse effects from what they believed to be ultrasound in public places (the ‘‘symptomatic’’ group), from subjects who had not (the ‘‘asymptomatic’’ group). I feel it is important to explain how Fletcher et al introduced the innovation of separating (the ‘‘symptomatic’’ group from the ‘‘asymptomatic’’ group: this is a good experimental way of overcoming the wide range in susceptibilities seen in the population, especially when the number of subject tested (as in the submitted manuscript) is small. Fletcher et al. (2018a) showed statistically significance for 'annoyance' and 'inability to concentrate', but was statistically proven by Fletcher et al. (2018a) and illustrated in a real-world case studies by Leighton (2020) and Leighton et al., (2020c) and Ueda (2014). Fletcher et al. (2018a) also found statistically significant galvanic skin responses (GSRs) in the ‘symptomatic’ group. Maccà et al. (2015) showed a statistically significant increases in asthenia (loss or lack of bodily strength) and vertigo, and in recalculating their data, Leighton (2016) confirmed these findings but also showed a statistically significant increase in ‘tingling in the limbs’. It is vital the authors properly review these studies or the reader will miss the imperative nature of the need to research and publish in this area. The first line of the CONCLUSIONS needs to reflect this small but important body of statistically significant findings.

Furthermore the phrase “To the authors’ knowledge, this study is the first to demonstrate that air-conducted US is capable of producing changes in functional brain activity” is inappropriate. I think, at most, the authors showed evidence of this, but what they present falls short of a clear “demonstration”.

• (1) Author’s response: As outlined in the response above, by rewording large parts of the introduction, we attempted to better situate our study within the current discourse on US-related health effects. We evaluated the aforementioned studies and incorporated many of the findings in the revised version of the manuscript. In addition, we also rewrote large parts of the conclusion to make it clear to the reader that our results add to the existing literature on statistically significant effects of US exposure.

4. SENSITIVITY and CONTROLS

Surely it is very important that the techniques used in this paper fail to register an effect, when patients report hearing the sound. This suggests that both MEG (Fujioka et al. 2002) and fMRI (this paper) lack the sensitivity to be used as sensory response threshold detectors for high frequency acoustics and ultrasonic exposure. This point should be made here. And in the first line of the Discussion. The paper leaves open the questions over whether the same can be said of the EEG data of Nittono (2020) et al. it is reported that no differences were detected – was it statistically tested? Could they hear the signals? In all these studies, did these past subjects actually hear the signals, or could they have been responding to something else? Please clarify.

In the same paragraph, the authors are coy as to why they are repeating the work of Kühler et al. (2019). It is vital that they are honest here. Kühler et al. (2019) used an inappropriate control. In the light of this, the current paper should not repeat the conclusions of that paper (‘the authors also found no evidence for auditory processing at frequencies above 14 kHz’) as this paper has already been cited by those who read it to mean human could not hear high frequencies. The fact that the control was inappropriate, and the point discussed here that MEG and fMRI are insufficiently sensitive for threshold detection, make it misleading to include text that would lead readers to believe that use of MEG and fMRI prove humans do not detect these frequencies.

The authors in the conclusion, “Whereas previous studies established rather unspecific associations between US exposure and a number of subjective symptoms (such as ‘annoyance’ or ‘discomfort’)”. This is not true given the work of Fletcher et al (2018a and b) and Macca et al (with stats re-run by Leighton 2016).

• (1) Author’s response: Please see our response to the 1.1, where the issue of signal detection by means of fMRI, MEG and EEG is addressed in detail. 

5. DETAILED COMMENTS

Abstract line 1. The point has been established in the literature (Leighton, 2017, 2018; Scholkmann, 2019; van Wieringen and Glorieux, 2018; Lubner et al., 2020; Paxton et al., 2018; Fletcher et al., 2018a,b,c; Dolder et al, 2018; Duck and Leighton, 2018) that, because the third octave bands that is centered on 20 kHz, extends down to 17.8 kHz, all the guidelines for Maximum Permissible Levels (MPLs) set for 20 kHz, extend down to 17.8 kHz, making (by default) the lower limit of the ultrasonic band 17.8 kHz, not 20 kHz. Cieslak et al (2020, 2021) acknowledge the argument to set the lower limit of ultrasound at 17.8 kHz, but comply with German tradition of setting it at 16 kHz, Given the 17.8 kHz has logical reasoning behind it, and beings in line most of the previous guidelines (which could not reasonable state 20 kHz as the limit whilst regulating down to 17.8 kHz when setting levels for ultrasonics exposure) I recommend you use that in the definition.

• (1) Author’s response: Thank you very much for the input and the useful references. In fact, a similar concern regarding the arbitrariness of defining ultrasound as sound frequencies above 20 kHz has also been raised by Reviewer #1. We therefore changed our definition of US in the Abstract and the introduction (p1, line 3; p2, line 4) and emphasize that there is a growing consensus among authors of more recent publications in the field, to use 17.8 kHz as the low limit of the ultrasonic frequency band (p2, lines 17 – 19).

Page 16. “Participants were neutral regarding the effects of US during the first run, although verbal reports varied considerably (mean = 0.00, SD = 1.51, min = -4, max = +3).” Is this strictly accurate? It would imply all subjects were neutral. So you mean the average response came out as neutral but that only a minority of subjects actually had a neutral response? This point was not clear and greater clarity would be appreciated.

• (2) Author’s response: We agree that the wording used in the article to describe the results was misleading. We therefore suggest the following correction: “During the first run, stimulation affected participants very differently, but there was no trend towards a perceived positive or negative influence on performance (mean = 0.00, SD = 1.51, min = -4, max = +3)” (p16, lines 3 – 5).

6. NEED FOR FURTHER DISCUSSION on ‘BELOW THRESHOLD’

Some of the other results are surprising. The main one is that in the BTC condition, where the US-tone is presented at 10 dB below the previously measured threshold, most subjects said (correctly) they could detect the sound, while in the NTC condition where no sound was presented most said (again correctly) that they could not hear any sound. So there is an explained discrepancy as to why they could (seemingly) hear the sound during the “resting state scanning” but not when their threshold was assessed. The authors do not really discuss why this might be, though they rule out test-retest reliability of the hearing threshold which was better than +/- 5 dB. 

• (1) Author’s response: This result was indeed surprising and at this point we cannot provide a conclusive explanation for it. In the article, we provided 3 – in our view strong – arguments in favor of the interpretation that stimulation was indeed below the HT. A) Sounds during (n-)BTC were administered by reducing the SPL by 10 dB (compared to a 5 dB increase for ATC). B) Test-retest reliability was checked for a number of participants, and C) the scanner was running during image acquisition while having been switched off during HT assessments (which may have increased HTs rather than decreasing them). However, as mentioned in the discussion, we cannot rule out that subtle changes in head- or earplug positioning may have affected sound transmission in the ear canal. 

• In the revised version of the manuscript, we try to address this topic in more detail and re-wrote the relevant segment of the discussion in the following way: “Given the lack of evidence for primary sensory processing anywhere in the brain, the question remains how US below the HT could have caused these changes in the first place. While little is known about the processing of subliminal auditory stimuli, it is well documented that changes in neuronal activity can be detected in response to visual stimuli presented below the perceptual threshold (e.g., see Brooks et al., 2012; Meneguzzo et al., 2014). In addition, the fact that IFG has also been implicated in contrast-enhancement for cases of low auditory discriminability (Alho et al., 1994; Opitz et al., 2002; Doeller et al., 2003) and that auditory change detection can be recorded for consciously imperceptible stimulus differences (Allen et al. 2000; Paavilainen et al., 2007) suggests that auditory stimulation does not necessarily have to produce a clear and distinct hearing impression in order to affect the CNS. Importantly, a similar conclusion can be drawn from Ascone et al.‘s study (2021), who showed that prolonged exposure to inaudible US at SPLs well below the HT was associated with significant grey matter volume reductions in large frontal clusters (including bilateral IFG), even though participants were asleep during most of the exposure. Apart from that, it remains unclear why 9 out of 10 participants who reported hearing the stimuli during ATC also had a hearing impression during BTC, while reporting correctly that no stimulation had occurred when the sound source was switched off (NTC). In general, the fact that the intra-individual spread of HTs in the five participants who completed the HT assessment twice on two non-consecutive days only ranged between -1 and +5 dB SPL, indicates that the experimental setup operated with enough precision to reliably present sound above the HT and that stimulation during (n-)BTC (presented 10 dB below the HT) was low enough to account for day-to-day fluctuations in hearing acuity. In addition, it needs to be emphasized that the presence of scanner noise increases HTs rather than decreases them (Röhl & Uppenkamp, 2012) and that changes in head- or earplug positioning would likely reduce the SPL in the ear canal, since the ear plug position was optimized via a 8 kHz probe tone prior to the scan. We can therefore only speculate that participants may have found it implausible that we did not expose them to US when asking for a perceptual report and therefore guessed randomly (with a rate of 66% false estimates, which is close to a 50% probability of guessing) (p20, lines 7 – 33).

Also, in the first of the three n-back scan runs, half the subjects did not say they could hear any tones, even when there were quite a few presented at 5 dB above threshold. But then in the next two cases they (mostly) could hear the sounds (as expected). It is not clear whether this anomaly can be explained due by differences in the test set up between threshold measurements and scanning, or some odd effect of the order of testing (I don't know what that might be). There needs to be clarity on this.

• (2) Author’s response: The three n-back runs were conducted without any interruptions (apart from the questions concerning the experience of the participants) or repositioning of the participants in the scanner, so apart from a minimal slippage of the ear plug due to head motion, which could have affected signal transmission in the ear canal, variability of the experimental conditions was kept to a minimum. Since no participant reported that the ear plug had actually slipped (and participants were aware of what this would have felt like, since a lot of time was spent finding the optimal fit of the ear plug before the experiments begun) we are quite confident that this anomaly cannot be explained by changes in experimental conditions. 

• Instead, we suggest the following explanation: “Moreover, when analyzing the verbal reports obtained after the n-back runs, it becomes evident that as the number of participants who reported hearing US during n-back increased from run 1 (7 out of 15) to run 3 (11 out of 15), the perceived influence of stimulation on task performance also changed from neutral in run 1 (mean = 0,00) to slightly negative (mean = 0,80) in run 3. One could thus speculate that as participants became more experienced with task performance, they also became more susceptible to attentional shifts induced by US (and thus aware of stimulation and its effect). (p19, lines 17 – 23). 

The authors appear to see differences in brain activity (bilateral IFG) between ATC and BTC stimulation, but it is not possible to tell from the paper whether they also see a difference between BTC and NTC (i.e. below threshold and no stimulus conditions) or ATC vs. NTC because they are unclear on this. The authors appear to claim this shows that the BTC condition is causing some effect in the brain, but it is currently impossible for the reader to tell whether this is valid. The authors found a difference in IFG between ATC vs BTC, but could it be that this was an inverse effect that was caused by the ATC case (relative to NTC)? I would have thought that the obvious hypothesis to test would be that the BTC condition gives the same response as the NTC condition (since in both cases there is no audible US).

• (3) Author’s response: To avoid overloading the article with data, we initially decided not to include the results for the other pairwise contrasts, although we also found a trend toward higher activation during n-BTC compared to n-NTC (p = 0.068). Since this trend can be considered as additional support for our finding that US below the HT can alter functional brain activity, we now included this result in the revised version of the manuscript (p15, lines 9 - 10).

So, when talking about the “below threshold” condition, the authors need to be careful. Indeed, I think it is not correct to call this ‘below threshold’ without qualification – because the reader will read the paper thinking this refers to levels below the threshold for inciting a response, but these ‘below threshold’ levels clearly are not always below threshold! Options might be that the detection sensitivity of the experiment is not sufficient. Or there might be large variations in the level reaching the ear canal based on fittings or movement of the fittings within in the ear canal or something like that because the sound is at such a high frequency. Given how much text the authors devote to the DISCUSSSION, they really don’t discuss this major issue much. 

• (4) Author’s response: We would like to refer the reviewer to our response to 6.1.

There may also be a problem with the correlations because as I understand it the authors only take a pleasantness rating when the person reports being able to perceive the sounds. It’s odd to read because the sound is “below threshold” yet the subjects only rate it’s pleasantness when they perceive it… if they can perceive it, then it’s obviously not below threshold … This identifies one limitation of the experiment vs the real world, which is that because of their rating scales (getting subjects to rate pleasantness and influence of the sounds), the authors are really getting the participants to focus in on the sounds each time, which they likely wouldn’t necessarily be doing with real-world exposure. Would it not be a valid criticism that these neural responses are quite sensitive to whether the subject is paying attention? If so, this needs discussion.

• (5) Author’s response: We certainly agree that the neural responses are very sensitive to whether the participant is paying attention, which is why we tried to emphasize the critical role of IFG in attentional processing at various points of the discussion. Importantly, at no point during the actual experiments were participants explicitly instructed to pay attention to the stimuli. During resting-state, participants were asked to lie in the scanner with eyes closed, not thinking of anything in particular, in order to simulate conditions more similar to those in the real world (p4, lines 32; p5 lines 5 – 6). This corresponds to the ”general instruction” given prior to a resting-state brain scan and is also found in other fMRI literature. Performing the “3-back version” of the n-back task is challenging and requires a great deal of sustained attention. This is also part of the reason, why we argued that IFG activation could reflect the triggering of attentional systems in response to potentially distracting auditory stimulation. Once participants are distracted and attention is allocated away from the task, this should be detectable as a functional change in the cognitive control network underlying successful task performance. In order to stress the important role of attention when aiming to identify the auditory centers for US perception, we also added the following segment in the discussion: 

• “Apart from that, it is known that mapping the tonotopic organization of PAC by means of fMRI is complicated by a number of additional factors (Langers et al., 2012). For example, the functional architecture of PAC exhibits considerable intersubjective variability (Rademacher et al. 2001) and several studies have also shown that neuronal response properties can be modulated via top-down attentional mechanisms (Bidet-Caulet et al. 2007; Woods et al. 2009; Paltoglou et al. 2011). It thus appears that future studies aimed at identifying the neural correlates of US perception would benefit from carefully considering these mechanisms, i.e. whether stimuli are actively being attended or just listened to passively, while also addressing intraindividual variability, f.e. by screening for US sensitivity as part of the recruitment process.“ (p21, lines 15 – 23).

7. STATISTICS

Also, some of the statistical analysis is odd, because in some places they appear to use all 15 subjects, when there were only at most 10 who should have had valid results (and possibly fewer as they could only include people who “heard” the tone in both the ATC and BTC conditions). The authors should revisit this point and explain themselves, or correct the analysis if necessary. The authors conduct quite a few statistical tests, where it is not clear whether these were planned before hand or not; this could inflate their false-positive rate if they were unplanned. These are definitely areas where the authors need to provide clarification.

• (1) Author’s response: Most of the analysis was pre-planned and conducted in order to best address the main hypotheses outlined in the introduction (p5, lines 11 – 18). After obtaining our main finding that IFG activation was significantly stronger in the contrast n-BTC vs. n-ATC, we looked for meaningful correlations between brain activation and perceptual / performance measures (both specified prior to data analysis). We believe that the number of statistical tests used in the study was comparatively small, since we only found one main imaging effect that we then correlated with these measures. 

• As part of the main analysis, we did not exclude individual data sets based on the fact whether a given participant did or did not (correctly or incorrectly) perceive a stimulus. The main goal of the study was to objectively analyze functional brain activity in response to carefully controlled US stimulus application as well as during cognitive task performance. In this respect, we believe that imaging data of all 15 participants was equally suitable for analysis. However, we repeated our analysis, restricted to only those participants who reported a hearing impression, when analyzing, whether US caused activation in one particular region of interest, i.e. primary auditory cortex (since we found no effect when including all 15 participants) (p15, lines 2 – 6). This was an additionally introduced test which could not be planned before because it was a constructive response to unexpected results of the study. Apart from that, we are not aware of any other parts of the fMRI analysis that would benefit from a more restrictive approach, despite the reduction in statistical power that would come along with it. 

• For the correlations between imaging data and perceptual reports, we also used results from all 15 participants, but those 4 participants who were not asked about their pleasantness rating, received a score of 0 (min = -5 = extremely unpleasant and max = +5 = extremely pleasant) (see 8.15. for further information). Moreover, as outlined in 8.12., the statistical tests for the direct comparison between pleasantness ratings of ATC and BTC (p15, lines 27 – 29) as well as the correlation pleasantness-ratings and error-rates now only involved those participants who reported a hearing impression during both conditions (and one participants reporting ‘unsure‘) (p16, lines 17 – 19).

8. SPECIFIC COMMENTS 

Please amend the paper to address these, rather than keep the paper barely changed and explain back to the reviewer, because if the reviewer was unclear what was meant, the average reader might well be, and they will not see any of the clarifying material in the response to reviewer. I would have thought the Editor might comment on whether it is appropriate to have so many acronyms in the Abstract. (Note from me regarding this comment: The number of acronyms/initialisms used in the abstract may be addressed in the final revision stage of the manuscript, if the paper is deemed suitable for publication. It would be wise to consider this issue in this revision to reduce your workload later).

• (1) Author’s response: To ensure a smoother reading, we removed the acronyms ‘CCN‘ for cognitive control network, and ‘ROI‘ for region of interest in the Abstract. 

p9. “HTs for pure tones with frequencies ranging from 8 to 26.5 kHz were measured monaurally (right ear) while participants lay in the scanner, in order to ensure that US audibility was preserved during the imaging process (background noise level was above the limit according to ISO 8253-1).” Were the HTs measured in an identical environment to the resting-state scanning condition, in terms of scanner noise etc? If so, why was the stimulus duration 1 sec during HT assessment and 2.75 sec during resting-state fMRI?

• (2) Author’s response: Thank you for the question. During the fMRI runs, repeated bursts of US with a frequency of 21.5 kHz, a duration of 2550 ms and on- and offset ramps of 100 ms were presented either at 5 dB above (‘above-threshold condition’; ATC) or 10 dB below the participants’ HT (‘below-threshold condition’; BTC). For the HT assessment, we reasoned that stimulus duration itself did not really matter, as long as it was long enough to elicit a clear and distinct auditory perception. We therefore chose sounds with similar characteristics, but with a duration of only 1000 ms. Apart from that, this also sped up the HT assessment, so participants had to spend less time in the scanner. 

p.10 “Participants were not informed about the order in which the runs were conducted.” Were the testers aware of the order – i.e. double blind?

• (3) Author’s response: This part of the study was conducted in a single blind fashion, as the order of stimulus conditions was changed between participants (so knowing the initial order, each of the following recordings was predictable) and the experimenter had to manually select the respective audio file. Due to the fact, that our post-recording questionnaire was very simplistic and standardized, we expect experimenter effects to be minimal. We added the following sentence to the revised manuscript: “The runs were conducted in a single blind fashion, i.e. only the experimenter knew the order of the stimulus conditions” (p10, lines 6 – 8).

p. 10 “whereas the ATC and BTC runs were counterbalanced across participants.”It could not be fully counterbalanced as there were an odd number of subjects. But admittedly any order effect is likely to be minor with at 8/7 imbalance. However the exact wording here cannot be correct.

• (4) Author’s response: We changed the wording in the following way: „Resting-state data acquisition always started with the NTC run, whereas the order of the stimulated runs was alternated across participants. Analysis of the resting-state data included 8 data sets in which ATC was followed by BTC, and 7 in which the order was reversed“ (p10, lines 5 – 6).

p.11 “In each run, the 12 blocks were aligned with the stimulus protocol in the following order: n-NTC, n-BTC, n-ATC, n-NTC, n-BTC, n-ATC, n-ATC, n-BTC, n-NTC, n-ATC, n-BTC, n-NTC”. What was the rationale for choosing this sequence?

• (5) Author’s response: We added the following information to the revised version of the manuscript. “This sequence was chosen in order to minimize the risk of stimulation order affecting task performance and/or brain activation (by alternating ATC and BTC) as well as the risk of data acquisition being influenced by whether a stimulation run was followed by a run without stimulation (by starting with NTC in the first 2 triplets and ending with NTC in the last 2 triplets) within a given amount of scanning time. In addition, after each block, a fixation period of 20 s was inserted, during which participants were looking at a black cross in the middle of the screen.” (p10, lines 28 – 33).

p.11 “After each of the six runs, participants were asked ’Did you hear sounds during the last run?’”. Given that the scanner noise is loud, presumably they were also instructed to only answer “yes” for sounds other than the scanner noise.

• (6) Author’s response: This is correct. Participants were aware of the importance of being able to distinguish scanner noise from the actual stimulus, which is why the category "unsure" was introduced. Since this hasn’t been clarified sufficiently in the article, we added the following information to the revised manuscript: “After each of the six runs, participants were asked ’Did you hear sounds during the last run?’ and asked to answer with ‘yes’, ’no’ or ’unsure’, if participants could not reliably distinguish between scanner noise and stimulus” (p11, lines 7 – 10).

p.14. “we contrasted whole brain data gathered during n-NTC vs. fixation”. I do not know what “fixation” means here. I am guessing it is results of a pre-task scan? Please clarify in the paper.

• (7) Author’s response: ‘Fixation’ refers to the period of 20 s after each n-block, during which participants were looking at a black cross in the middle of the screen. This information has been added to the revised version of the manuscript (p10, lines 31 – 32) and Fig. 3 has been updated accordingly.

p. 15 “To do so, we calculated pairwise contrasts between activations during the two stimulus conditions (n-ATC + n-BTC) and n-NTC, however no significant differences were found (p < 0.001, uncorrected)”. Does “n-ATC + n-BTC”. Does this mean that the two activations were first summed, and then contrasted with n-NTC, or that two contrasts were conducted: n-ATC vs n-NTC and n-BTC vs n-NTC? Further down, it seems that excitation of IFG for n-BTC > n-ATC (statistically significant). What about n-BTC vs n-NTC? This seems like a sensible planned comparison, given the hypothesis that “BTC cannot be detected and so would not be expected to lead to excitation? If the BTC excitation is causing some effect in the brain, does this show up in the BTC vs. NTC contrast?

• (8) Author’s response: The phrasing used in the article is indeed misleading. Pairwise contrasts were calculated between each stimulus condition and the no-tone condition separately, i.e. n-ATC vs. n-NTC and n-BTC vs. n-NTC (p15, line 1 – 4). Regarding the 2nd point, we would like to refer the reviewer to our response to 6.3. However, it is important to mention that our hypothesis was not that “BTC cannot be detected and so would not be expected to lead to excitation”. In the introduction we stated our main hypothesis without making specific predictions regarding the effects of US below or above the HT (see p5, line 23 – 28). In fact, we assumed it was entirely possible that both types of stimulation could have an effect on functional connectivity (and perhaps in a different way).

p. 15 “To do so, we calculated pairwise contrasts between activations during the two stimulus conditions (n-ATC + n-BTC) and n-NTC, however no significant differences were found (p < 0.001, uncorrected)”. Presumably this should be p>0.001 if it was not significant?

• (9) Author’s response: In neuroimaging publications, it is customary to state the threshold of the statistical testing. To make this clear, we changed wording in brackets to: “thresholded at p < 0.001, uncorrected” (p15, line 3 – 4).

p.15 “However, when contrasting the two stimulation runs directly (n-BTC vs. n-ATC) we found a strong activation in bilateral inferior frontal gyrus (IFG, triangular part) only when US was presented below the HT (p < 0.001, cluster > 30)”. I don’t know what “cluster >30” means here, but it probably makes sense to an fMRI expert. Please clarify for the acousticians and audiologists who will be reading this.

• (10) Author’s response: The following information was added to the revised manuscript: “Typical fMRI analyses include between 100000 to 200000 voxels resulting in numerous statistical tests, which must be appropriately corrected for multiple comparisons. Instead of testing each voxel individually, most fMRI analyses test, whether a given cluster of voxels exhibits statistically significant activation, assuming that activations in proximate voxels are not fully independent. To display the results of the group analysis, statistical values were thresholded with a level of significance of p < 0.001 (z > 3.09, uncorrected); a significant effect was reported when the volume of the cluster was greater than the minimum cluster size determined by Monte Carlo simulation above which the probability of type I error was < 0.05.” (p13, line 4 – 12).

p. 15. “after NTC, 11 out of 15 accurately stated that no stimulation had taken place, two were unsure and two reported that sounds had been presented.” Not consistent with table 2 which states 3 were unsure.

• (11) Author’s response: This has been corrected accordingly. The information provided in table 2 is correct.

p. 15. “Apart from that, paired t-test (two-tailed) revealed that stimulation during ATC (mean = -1.00, SD = 1.13) was rated significantly more unpleasant compared to BTC (mean = 0.07, SD = 0.70) (t(15) = 3.23, p = 0.006).” How are there get 15 degrees of freedom in the t-statistic? If all 15 were included then you would have 14 dfs. But presumably only subjects who reported sounds in both ATC and BTC conditions who could be included – so at most 10, and possibly fewer, because it is stated that only subjects who reported sounds present then rated the pleasantness of the sound. And it could be <10, since the paired test can only be performed on subjects who were correct in both ATC and BTC conditions. It isn’t stated how many of the 10 who reported sounds present in ATC also reported them present in BTC. And also how many of these wrongly reported sounds present in NTC?

• (12) Author’s response: Thank you for the remark. You are correct that the paired t-test can only be performed on participants who had a hearing impression in both ATC and BTC conditions. This leaves us with a total 10 participants (9 who reported ´yes´ twice and 1 who reported ´unsure´), hence dfs = 9. When re-analysing our data, we obtained slightly different values, yet the overall effect stayed the same. We corrected this in the revised manuscript in the following way: “Here, paired t-test (two-tailed) revealed that stimulation during ATC (mean = -1.30, SD = 1.160) was rated significantly more unpleasant compared to BTC (mean = 0.20, SD = 0.789) (t(9) = -3.545, p = 0.006)” (p15, lines 29 – 32; Tab 2). Regarding the second to last point raised: In the initial draft of the manuscript, we already mention that 9 out of 10 participants who reported hearing sound during ATC, also had a hearing impression during BTC (p20, lines 20 – 23).

p. 15. “When participants were asked ‘Did you hear sounds during the last run?’ after the first n-back run, 7 out of 15 (one ‘unsure’)”. Since the run contained at four blocks ATC, why do you think the sounds were not detected?

• (13) Author’s response: We would like to refer the reviewer to our response to 6.2. 

p. 16. “While no correlations between resting-state brain data and verbal reports reached significance, we found that bilateral IFG activation during n-BTC vs. n-ATC was associated with verbal reports after resting-state in two ways: First, we found that the more pleasant a tone was perceived during ATC, the higher activation in bilateral IFG during ATC was (r = 0.602, p = 0.023).” Should that be “higher activation in bilateral IFG during n-ATC vs n-BTC”?

• (14) Author’s response: Thank you again for the very careful reading. The correct phrasing should be “higher activation in bilateral IFG during n-BTC vs. n-ATC (p16, line 13 – 14).

p. 16. “While no correlations between resting-state brain data and verbal reports reached significance, we found that bilateral IFG activation during n-BTC vs. n-ATC was associated with verbal reports after resting-state in two ways: First, we found that the more pleasant a tone was perceived during ATC, the higher activation in bilateral IFG during ATC was (r = 0.602, p = 0.023).” What are the degrees of freedom for the estimate of r? For verbal reports in the resting state, there were only 10 subjects who were asked about pleasantness, so presumably there were only 9 dfs for this test. R=0.602 is not significant with dfs=9 on a two-tailed test.

• (15) Author’s response: This correlation was calculated with dfs = 14, since brain data gathered from all 15 participants was correlated with perceptual ratings. For those subjects who were not asked about their pleasantness rating, a score of 0 (min = -5 = extremely unpleasant and max = +5 = extremely pleasant) was used. In addition, while checking our calculations, we also found that the rating score for one participant during ATC was miscalculated when transforming the Likert scale from -5 – +5 to 0 – 10 . After correcting the error, we obtained slightly different r- and p-values: 1. The more pleasant a tone during ATC, the higher activation in bilateral IFG (r = 0.579, p = 0.024). 2. The more unpleasant a tone during BTC relative to ATC, the higher activation in bilateral IFG (r = 0.711, p = 0.003). 

p.16. A number of different correlations are presented. Were these all a priori planned correlations? ATC pleasantness vs. n-ATC minus n-BTC IFG activation is presented, and also ATC pleasantness minus BTC pleasantness vs. n-ATC minus n-BTC IFG activation. Why these combinations? E.g. why not also BTC pleasantness vs. n-ATC minus n-BTC IFG activation? Other correlation coefficients are calculated involving for IFG activation, RTs, pleasantness rating, error rates. This leads to a large number of potential correlations and hence an inflated type 1 error rate unless there were selected planned comparisons.

• (16) Author’s response: We would like to refer the reviewer to our response to 7.1.

p. 17. “First, we showed that the more unpleasant sound was perceived during BTC compared to ATC the higher signal strength in bilateral IFG was (although ATC was generally experienced as more uncomfortable).” In the discussion, the authors don’t mention that they report a correlations between pleasantness in ATC and IFG activation in ATC (though I think the latter might be n-ATC vs. n-BTC). Is this also consistent with your hypothesis?

• (17) Author’s response: We did not form any a prior hypothesis about specific correlation between perceptual ratings and specific brain activation effects. However, the fact the overall ATC was experienced as more unpleasant, was consistent with our hypothesis (p5, lines 23 – 25). We did not discuss the correlation between pleasantness in ATC and IFG activation explicitly, since the effect was much more pronounced when the difference scores (BTC vs. ATC) were correlated with the imaging data. Nevertheless, this result indicates that IFG activation does not encode for pleasantness or unpleasantness of a given stimulus per se. In the case of US, both the stimulation condition and the perceived effect appear to have a differential impact on IFG activation, with the strongest activation being measured when the discrepancy between the perceived influence of US at BTC and ACT is greatest.

P 17. “WM” not defined. Presumably “working memory”.

• (18) Author’s response: Correct, this has been changed accordingly (p17, line 24).

Ethics statement and page 7. “The study was conducted according to the Declaration of Helsinki with approval of the ethics committee of the German Psychological Association (DGPs).” The authors should state the year of the Declaration of Helsinki (because it changes significantly with different versions).

• (19) Author’s response: The following information has been added to the revised manuscript: The study was conducted according to the Declaration of Helsinki (64th WMA General Assembly, 2013) (p6, line 9).

REFERENCES

• Cieslak M, Kling C and Wolff A (2020) Ultrasound exposure in a workplace and a potential way to improve its measurement methodology. 2020 IEEE International Workshop on Metrology for Industry 4.0 & IoT, 2020, pp. 172-176, doi: 10.1109/MetroInd4.0IoT48571.2020.9138223.

• Cieslak M, Kling C, Wolff A. (2021) Development of a Personal Ultrasound Exposimeter for Occupational Health Monitoring. International Journal of Environmental Research and Public Health. 18(24):13289. https://doi.org/10.3390/ijerph182413289

• Duck, F. and Leighton, T.G. (2018) Frequency bands for ultrasound, suitable for the consideration of its health effects. J. Acoust. Soc. Am. 144(4) 2490-2500 (doi: 10.1121/1.5063578)

• Dolder, C.N., Fletcher, M.D., Lloyd Jones, S., Lineton, B., Dennison, S.R., Symmonds, M., White, P.R. and Leighton, T.G. (2018) Measurements of ultrasonic deterrents and an acoustically branded hairdryer: Ambiguities in guideline compliance. J. Acoust. Soc. Am. 144(4), 2565-2574 (doi: 10.1121/1.5064279)

• Fletcher, M.D., Lloyd Jones, S., White, P.R., Dolder, C.N., Leighton, T.G. and Lineton, B. (2018a) Effects of very high-frequency sound and ultrasound on humans. Part I: Adverse symptoms after exposure to audible very-high frequency sound. J. Acoust. Soc. Am. 144(4), 2511-2520 (doi: 10.1121/1.5063819)

• Fletcher, M.D., Lloyd Jones, S., White, P.R., Dolder, C.N., Leighton, T.G. and Lineton, B. (2018b) Effects of very high-frequency sound and ultrasound on humans. Part II: A double-blind randomized provocation study of inaudible 20-kHz ultrasound. J. Acoust. Soc. Am. 144(4), 2521-2531 (doi: 10.1121/1.5063818)

• Fletcher, M.D., Lloyd Jones, S., White, P.R., Dolder, C.N., Lineton, B. and Leighton, T.G. (2018c) Public exposure to ultrasound and very high-frequency sound in air. J. Acoust. Soc. Am. 144(4), 2554-2564 (doi: 10.1121/1.5063817)

• Leighton, T.G., (2017). Comment on ‘Are some people suffering as a result of increasing mass exposure of the public to ultrasound in air? Proc. Math. Phys. Eng. Sci. 473, 20160828

• Leighton, T.G. (2018) Ultrasound in air - Guidelines, applications, public exposures, and claims of attacks in Cuba and China. J. Acoust. Soc. Am. 144(4) 2473-2489 (doi: 10.1121/1.5063351).

• Leighton, T. G. (2020) Ultrasound in air - Experimental studies of the underlying physics are difficult when the only sensors reporting contemporaneous data are human beings. Physics Today, 73(12), 39-43 (doi: 10.1063/PT.3.4634).

• Leighton, T. G., Currie, H. A. L., Holgate, A., Dolder, C. N., Lloyd Jones, S., White, P. R. and Kemp, P. S. (2020a). Analogies in contextualizing human response to airborne ultrasound and fish response to acoustic noise and deterrents, Proceedings of Meetings on Acoustics (POMA) 37, 010014 (doi: 10.1121/2.0001260).

• Leighton, T. G., Lineton, B., Dolder, C. N. and Fletcher, M. D. (2020b) Public Exposure to airborne ultrasound and Very High Frequency sound. Acoustics Today, 16(3), 17-26 (doi: 10.1121/AT.2020.16.3.17).

• Lubner R. J., Kondamuri N. S., Knoll R. M., Ward B. K., Littlefield P. D., Rodgers D., Abdullah K. G., Remenschneider A. K. and Kozin E. D. (2020) Review of Audiovestibular Symptoms Following Exposure to Acoustic and Electromagnetic Energy Outside Conventional Human Hearing. Frontiers in Neurology, 11, Article 234, DOI=10.3389/fneur.2020.00234

• Maccà, I., Scapellato, M. L., Carrieri, M., Maso, S., Trevisan, A. and Bartolucci, G. B. (2015) High-frequency hearing thresholds: effects of age, occupational ultrasound and noise exposure. Int. Arch. Occup. Environ. Health 88, 197–211. (doi:10.1007/s00420-014-0951-8)

• Mapp, P. A. (2018) Potential audibility of ultrasonic signal monitoring of Public Address and Life Safety Sound Systems, J. Acoust. Soc. Am. 144(4), 2539–2547

• Scholkmann, F. (2019) Exposure to High-Frequency Sound and Ultrasound in Public Places: Examples from Zurich, Switzerland. Acoustics 1, 816 (2019).

• Ueda, M., Ota, A. and Takahashi, H. (2014) Investigation on high-frequency noise in public space. Internoise 2014, Melbourne Australia, 16–19 November 2014, 7 p.

• Ullisch-Nelken C., Wolff, A., Schöneweiß, R. and Kling, C. (2017), A measurement procedure for the assessment of industrial ultrasonic noise, Proceedings of the 25th International Congress on Sound and Vibration, Vol. 4, pp. 2433–2438, Curran Associates, Red Hook, NY, USA

• van Wieringen, A. and Glorieux, C. (2018) Assessment of short-term exposure to an ultrasonic rodent repellent device,” J. Acoust. Soc. Am. 144(4), 2501–2510.

---

## [Decision Letter · Decision Letter 1]

3 Nov 2022

Air-conducted ultrasound below the hearing threshold elicits functional changes in the cognitive control network.

PONE-D-21-19549R1

Dear Dr. Weichenberger,

We’re pleased to inform you that your manuscript has been judged scientifically suitable for publication and will be formally accepted for publication once it meets all outstanding technical requirements.  I would like to ask you reconsider making your data publicly available, as the PLOS One standard and goal is open, transparent communication of scientific information.  Posting your data without subject identifiers would be ideal.  Making your data available upon request is less ideal.  Please reconsider this option as you finalize your manuscript.

Kind regards,

Brenton G. Cooper, Ph.D.

Academic Editor

PLOS ONE

Additional Editor Comments (optional):

Reviewers' comments:

Reviewer's Responses to Questions

**Comments to the Author**

1. If the authors have adequately addressed your comments raised in a previous round of review and you feel that this manuscript is now acceptable for publication, you may indicate that here to bypass the “Comments to the Author” section, enter your conflict of interest statement in the “Confidential to Editor” section, and submit your "Accept" recommendation.

Reviewer #1: All comments have been addressed

Reviewer #2: All comments have been addressed

2. Is the manuscript technically sound, and do the data support the conclusions?

Reviewer #1: Yes

Reviewer #2: Yes

3. Has the statistical analysis been performed appropriately and rigorously? 

Reviewer #1: Yes

Reviewer #2: Yes

4. Have the authors made all data underlying the findings in their manuscript fully available?

Reviewer #1: Yes

Reviewer #2: No

5. Is the manuscript presented in an intelligible fashion and written in standard English?

Reviewer #1: Yes

Reviewer #2: Yes

6. Review Comments to the Author

Reviewer #1: (No Response)

Reviewer #2: I commend the authors on an excellent job. They have taken great care. I read the comments from Reviewer 1, too, and the responses to that, and was impressed by the care the authors took in addressing both sets of comments.

7. PLOS authors have the option to publish the peer review history of their article (what does this mean?). If published, this will include your full peer review and any attached files.

Reviewer #1: No

Reviewer #2: No

---

## [Editor Report · Acceptance letter]

1 Dec 2022

PONE-D-21-19549R1 

Air-conducted ultrasound below the hearing threshold elicits functional changes in the cognitive control network 

Dear Dr. Weichenberger:

I'm pleased to inform you that your manuscript has been deemed suitable for publication in PLOS ONE. Congratulations! Your manuscript is now with our production department. 

Kind regards, 

on behalf of

Dr. Brenton G. Cooper 

Academic Editor

PLOS ONE